# Genome information processing by the INO80 chromatin remodeler positions nucleosomes

Elisa Oberbeckmann [1,7,9], Nils Krietenstein[1,8,9], Vanessa Niebauer [2,3], Yingfei Wang[4], Kevin Schall[2,3], Manuela Moldt[2,3], Tobias Straub [5], Remo Rohs [4], Karl-Peter Hopfner [2,3✉], Philipp Korber[1✉] & Sebastian Eustermann [2,3,6✉]

The fundamental molecular determinants by which ATP-dependent chromatin remodelers organize nucleosomes across eukaryotic genomes remain largely elusive. Here, chromatin reconstitutions on physiological, whole-genome templates reveal how remodelers read and translate genomic information into nucleosome positions. Using the yeast genome and the multi-subunit INO80 remodeler as a paradigm, we identify DNA shape/mechanics encoded signature motifs as sufficient for nucleosome positioning and distinct from known DNA sequence preferences of histones. INO80 processes such information through an allosteric interplay between its core- and Arp8-modules that probes mechanical properties of nucleosomal and linker DNA. At promoters, INO80 integrates this readout of DNA shape/mechanics with a readout of co-evolved sequence motifs via interaction with general regulatory factors bound to these motifs. Our findings establish a molecular mechanism for robust and yet adjustable +1 nucleosome positioning and, more generally, remodelers as information processing hubs that enable active organization and allosteric regulation of the first level of chromatin.

[1] Biomedical Center (BMC), Faculty of Medicine, Ludwig-Maximilians-Universität München, Martinsried, Germany. [2] Gene Center, Ludwig-Maximilians-Universität München, Munich, Germany. [3] Department of Biochemistry, Faculty of Chemistry and Pharmacy, Ludwig-Maximilians-Universität München, Munich, Germany. [4] Departments of Quantitative and Computational Biology, Chemistry, Physics and Astronomy, and Computer Science, University of Southern California, Los Angeles, CA, USA. [5] Core Facility Bioinformatics, Biomedical Center, Faculty of Medicine, Ludwig-Maximilians-Universität München, Martinsried, Germany. [6] European Molecular Biology Laboratory (EMBL), Structural and Computational Biology Unit, Heidelberg, Germany. [7] Present address: Department of Molecular Biology, Max Planck Institute for Biophysical Chemistry, Göttingen, Germany. [8] Present address: Department of Biochemistry and Molecular Pharmacology, University of Massachusetts Medical School, Worcester, MA, USA. [9] These authors contributed equally: Elisa Oberbeckmann, Nils Krietenstein. ✉email: hopfner@genzentrum.lmu.de; pkorber@lmu.de; sebastian.eustermann@embl.de

The packaging of DNA with histones into nucleosomes underpins the maintenance and regulation of genome information in eukaryotes[1,2]. Genome-wide mapping of chromatin revealed highly defined patterns of nucleosomes carrying a combinatorial landscape of histone variants and modifications[3–8]. These patterns entail well-positioned nucleosomes, which occupy the same genomic position across a cell population and even adopt equivalent positions relative to genomic sites of equivalent function like transcription start sites (TSS)[6,7]. Most prominently, nucleosome-depleted regions (NDRs) at promoters of active or poised genes are flanked by a well-positioned hallmark nucleosome (+1 nucleosome) that is the first in a regular nucleosome array over the transcribed region[9]. These stereotypic NDR-array patterns are conserved from yeast to man, and changes within their configuration play a pivotal role in transcriptional regulation, e.g., during cell differentiation and stress response[10,11]. Understanding the fundamental molecular determinants of nucleosome positioning is likely to reveal core principles by which genome regulation occurs.

A nucleosome position is defined by the DNA sequence that is wrapped around the histone octamer[12]. While this DNA sequence always answers the question, Where is this nucleosome?, it may, but need not, answer the question, How was the nucleosome placed there?. Histone octamers may form nucleosomes virtually at any DNA sequence position in the genome[13]. A molecular mechanism that consistently places a nucleosome at a particular genome position across a cell population must select this position against competing positions. This selection may be based on genetic information encoded within DNA sequence or on epigenetic information like histone modifications and variants or other chromatin-associated factors. Regarding DNA sequence information, pioneering studies proposed two mechanisms (Fig. 1a). One mechanism, designated as a genomic code for nucleosome positioning, relies on the intrinsic preference of nucleosomes to assemble on DNA sequences that favor wrapping around the histone octamer[14,15]. In this case, the nucleosomal DNA-histone interaction directly determines the position. The other mechanism, designated as statistical positioning of nucleosomes, requires DNA sequence-specific binding of a barrier factor, to which one or several nucleosomes are aligned

regardless of the octamer-bound DNA sequences[16]. The principal difference between these two mechanisms illustrates two extremes, which pertain to the central question whether DNA sequence information directly or indirectly determines a nucleosome position. If directly, the nucleosome positioning mechanism reads out the DNA sequence information at the resulting nucleosome position itself. If indirectly, DNA sequence is read somewhere else, and the resulting positioning information is relayed by alignment mechanisms that position nucleosomes relative to barriers and other nucleosomes. In the latter case, the DNA sequence bound by the histone octamer would define, but not directly determine, the genomic position of a nucleosome.

In recent years, it has become clear that the pure versions of these two mechanistic extremes fail to explain nucleosome positioning in vivo. Intrinsic histone octamer preferences, as operationally assessed by salt gradient dialysis (SGD) reconstitution from purified DNA and histones[13], cannot recapitulate NDR-array patterns in vitro[17,18], and internucleosomal distances (spacing) are independent of nucleosome density in vivo[19,20] and in vitro[18,21] in contrast to predictions of the statistical positioning mechanism[16,22].

Instead, ATP-dependent chromatin remodelers have now been established as decisive nucleosome positioning factors both in vivo and in vitro. Chromatin remodelers often form multi-subunit macromolecular complexes and are grouped into four families: INO80/SWR1, SWI/SNF, ISWI, and CHD. By using energy derived from ATP hydrolysis, remodelers alter histone-DNA interactions resulting in nucleosome translocation (sliding), ejection, and reconfiguration[23]. Mutations in genes encoding remodeler subunits, especially combined mutations, lead either to compromised nucleosome patterns and composition, or are lethal[20,24–28]. Complementary to genetic studies, cell-free reconstitutions provided direct evidence for the critical role of chromatin remodelers in nucleosome positioning and allowed to distinguish remodeler contributions from those of other factors, like the transcription and replication machinery[18,29]. Nucleosomes were assembled by SGD, even for an entire genome with yeast genomic DNA fragments or plasmid libraries[17,18,29,30]. The largely non-physiological nucleosome positions generated by SGD were turned in an ATP-dependent manner into in vivo-like

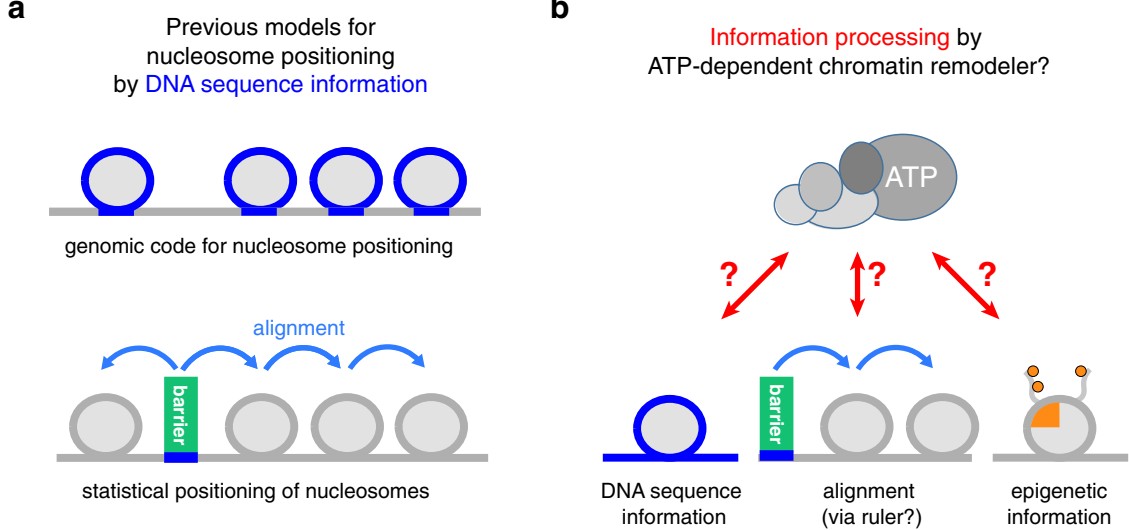

**Fig. 1 Models for nucleosome positioning mechanisms. a** Genomic code for nucleosome positioning[14,15] and statistical positioning[16] are two previous models, which exemplify a direct versus indirect role, respectively, of DNA sequence information (blue) for determining nucleosome positioning. **b** In light of the decisive role of ATP-dependent chromatin remodelers in nucleosome positioning[24,28,29,86], we asked if and how these large, macromolecular machines actively process (epi)genetic information together with their own remodeler-specific information into stable nucleosome positioning.

NDR-array patterns either by addition of whole cell extracts[18] or, remarkably, also by addition of purified yeast remodelers[29]. For example, addition of the yeast INO80 or the SWI/SNF-type RSC remodeling complex to SGD chromatin without other factors already generated hallmark features of in vivo-like nucleosome organization, i.e., +1 nucleosomes and broad NDRs at promoters, respectively[29]. This argued for a remodeler-mediated direct readout of positioning information, possibly involving DNA sequence features[29,31] and epigenetic information[23]. Notably, various remodelers contain reader domains of histone marks, while most of them lack classical sequence-specific DNA binding domains. This led to the proposal that remodelers may recognize sequence-dependent structural features of DNA such as DNA shape[29,32]. Ample and growing evidence in the context of transcription factors binding to DNA underscores the functional relevance of DNA shape features in genome regulation[33]. Such features may be relevant at poly(dA:dT)-rich promoter sequences, implicated in regulation of RSC activity during NDR formation[29,31], and we hypothesized that DNA shape may also play a role during +1 nucleosome positioning by INO80[29]. In contrast, other remodelers, such as the yeast ISW1a and ISW2 complexes, could not generate in vivo-like nucleosome positions on their own but required sequence readout by other factors. General regulatory factors (GRFs) are sequence-specific DNA binding proteins, often essential for viability and involved in transcription or replication regulation via their impact on chromatin organization[34–36]. Addition of purified GRFs, e.g., yeast Reb1 or Abf1, enabled the ISW1a and ISW2 remodelers to align regular nucleosome arrays relative to the GRF binding sites[29]. This argued in turn for remodeler-mediated indirect read out of positioning information via processive nucleosome alignment at GRFs bound to their specific sites as well as among nucleosomes, possibly involving a protein ruler[37].

Although cell-free reconstitution and genetic studies established the critical importance of remodelers in determining the genomic organization of nucleosomes, the dissection of the underlying molecular mechanism and the required information has proven difficult. Recent structural work shed light onto the architecture of different remodelers and how they might translocate mononucleosomes[38]. However, there remains the conundrum that the principal remodeler activity of mobilizing nucleosomes must be regulated such that it results in stable nucleosome positions relative to genomic sequence.

In this study, we directly addressed this fundamental conundrum by asking which kind of DNA sequence, histone, barrier, or other epigenetic information provides the required input, and how remodelers turn this information input into stable nucleosome positioning (Fig. 1b). We advanced whole-genome reconstitutions into a fully recombinant, de novo approach. In this system full biochemical control is established by using recombinant components in conjunction with high-resolution structural information enabling the identification of remodeling mechanisms. Not only the core mechanism of remodelers, as studied so far mainly in mononucleosome assays, but also the extended functions arising from remodeling of chromosomal multinucleosome substrates as well as the readout of physiological genomic DNA sequences and other nucleosome positioning information can be assessed at a detailed mechanistic level. We used the yeast genome and the multi-subunit structure of the INO80 complex as a paradigm to identify and probe the information and mechanism by which remodelers read information and translate it into stable nucleosome positions. In the accompanying study[39], we addressed how remodelers propagate nucleosome positioning information via an alignment mechanism so that phased and regular nucleosomal arrays are generated. Taken together, our data reveal that and how remodelers are

information processing hubs. Genome information encoded within DNA shape/mechanics as well as in DNA sequence motifs bound by barrier factors is actively read out by the remodelers and integrated via the allosteric interplay of their molecular machinery into nucleosome positions.

## Results

**A fully recombinant approach for de novo whole-genome reconstitutions**. To explore how ATP-dependent chromatin remodelers place nucleosomes at in vivo-like positions, we advanced whole-genome reconstitutions[18,29,30] into a fully recombinant de novo approach (Fig. 2a). We established recombinant production of highly active and stoichiometric INO80 complex (Supplementary Fig. 1a, b) and performed whole-genome reconstitutions using recombinant histones and a clonally defined and completely sequenced S. cerevisiae genomic plasmid library[40]. This leverages, compared to previously used ill-defined plasmid libraries, endogenous fly embryo histones and purifications of endogenous remodelers[29], the full potential of biochemical systems: (1) a fully defined 15-subunit S. cerevisiae INO80 complex, amendable for structure-guided mutagenesis, (2) histones without posttranslational modifications (PTMs) and amendable for mutagenesis, and (3) fully defined DNA templates for chromatin assembly. We used MNase-seq to measure resulting nucleosome positions.

**DNA sequence and globular histone octamer information is sufficient for in vivo-like +1 nucleosome positioning by INO80**. This recombinant system enabled us to identify the minimal information for nucleosome positioning by INO80. Consistent with its localization and function in vivo[41], INO80 positions in vivo-like +1 nucleosomes adjacent to NDRs (Fig. 2b[29]). As equally pronounced +1 nucleosome positioning activity was observed for recombinant as for endogenous INO80 (Fig. 2b, left), we concluded that no yeast-specific PTMs of INO80 were required and no co-purified yeast contaminant was responsible. To control the specificity of the highly pure INO80 complex (Supplementary Fig. 1a, b), we assayed an INO80 complex, which carries a Walker B motif mutation within its Ino80 ATPase motor protein (Supplementary Fig. 1c), and excluded that nucleosome positioning activity was due to any co-purifying factor(s) from insect cells. Intriguingly, our recombinant whole-genome reconstitutions established conditions, under which INO80 generated extensive nucleosome arrays (e.g., upon addition of Reb1, see below). This served as starting point for the study of nucleosome spacing mechanisms as described in the accompanying paper[39].

Next, we asked whether epigenetic information derived from histone modifications or variants was required for +1 nucleosome positioning. Histone variants, for example H2A.Z, may alter sequence-dependent DNA preferences of the histone octamer[42]. However, compared to SGD chromatin prepared with endogenous fly histones, using either recombinant human or yeast histones resulted in very similar nucleosome positioning by INO80 (Fig. 2b, right). Patterns were less pronounced with yeast histones, which we attributed to their known propensity to form less-stable nucleosomes[43]. As the species origin of the histones did not matter much, we went more minimalistic and asked if just the globular histone domains were sufficient. SGD chromatin with recombinant tailless human histones still allowed INO80 to position in vivo-like +1 nucleosome position (Fig. 2b, right). We observed increased sliding rates with tailless compared to full-length histone nucleosomes (Supplementary Fig. 1d) consistent with previous studies[44–46]. Nonetheless, this increased sliding rate

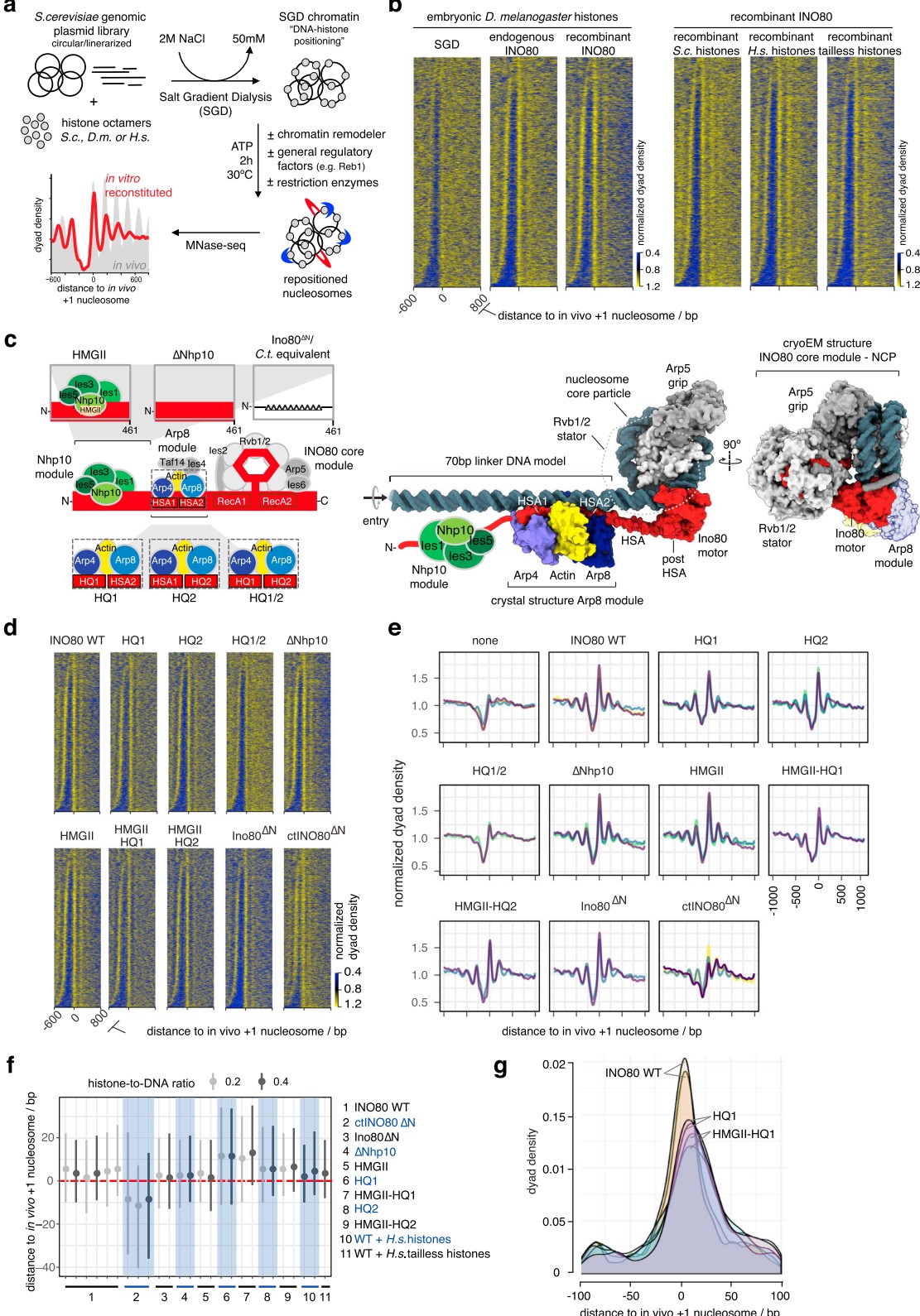

did not abrogate formation of the steady state nucleosome positioning pattern.

Taken together, we concluded that neither histone modifications nor histone variants nor histone tails nor yeast-specific INO80 modifications are absolutely required for INO80's principal activity to position in vivo-like +1 nucleosomes.

Consequently, INO80 can generate such positioning solely by processing information from genomic DNA sequences and the globular histone octamer. Nonetheless, a readout of epigenetic information by remodelers is expected to play a pivotal role in the regulation of nucleosome positioning, e.g., in response to changes in the cellular environment, as discussed further below.

**Fig. 2 Fully recombinant genome-wide reconstitution of nucleosome positioning by INO80. a** Overview of genome-wide in vitro chromatin reconstitution system. **b** Heat maps of MNase-seq data for SGD chromatin assembled with embryonic or recombinant histones from the indicated species (*H.s. Homo sapiens, S.c. Saccharomyces cerevisiae*) and remodeled with endogenous (GSM1855399)[29] or recombinant *S. cerevisiae* INO80 complex as indicated. Heat maps are aligned at in vivo +1 nucleosome positions and sorted by NDR length. Single replicates were plotted, see Supplementary Fig. 1c and Supplementary Data 1 for all replicates. **c** Left panel: schematic of INO80 complex submodule and subunit organization (middle) with zoom into Nhp10 (top) or Arp8 module (bottom) showing three mutant versions each. Right panel: composite model of INO80 based on high-resolution cryoEM structure of ctINO80 (*Chaetomium thermophilum* INO80) core in complex with a mononucleosome[44] and X-ray structure of Arp8 module[47] modeled on 70 bp linker DNA. The AAA+ ATPase hetero-hexamer Rvb1/2 (omitted for clarity and indicated by a dotted line in the left representation) acts as a stator for the Ino80 ATPase motor and the nucleosome gripping subunit Arp5. The direction of entry DNA translocation is indicated. **d** Heat maps of MNase-seq data of individual replicates for SGD chromatin incubated with the indicated recombinant WT (WT) or mutant INO80 complexes (as in **c**, left, and also with combinations of HMGII and HQ1 or HQ2 mutations) from *S. cerevisiae* or *C. thermophilum* (ctINO80^ΔN). **e** Composite plots of MNase-seq data of individual replicates for SGD chromatin incubated with the indicated recombinant WT (WT) or mutant INO80 complexes as in **d**. Each color represents an independent replicate (n = 3 for SGD ("none"), WT, HQ1, HQ2, ctINO80^ΔN; n = 2 for HQ1/2, HMGII-HQ1, HMGII-HQ2, Ino80^ΔN). Composite plots of replicates from **d** are shown as purple traces. **f** Distributions of distances between +1 nucleosome positions determined by paired-end sequencing after reconstitution by the indicated combinations of INO80 complexes and histones at the indicated histone-to-DNA mass ratio relative to in vivo +1 nucleosome positions. Dots mark the medians, vertical lines the interquartile distances. Alternating white and gray vertical zones group replicates of the indicated remodeler/histone combinations. **g** Density distributions of MNase-seq reads relative to in vivo +1 nucleosome positions of biological replicates with INO80 WT (yellow and orange area), HQ1 (pink areas), and HMGII-HQ1 (purple areas) mutant complexes as in **f**.

---

**Structure-based site-directed mutagenesis probes nucleosome positioning by INO80.** Having identified a minimal set of components, from which INO80 derives nucleosome positioning information, we set out to specify this information and to dissect the molecular mechanism, by which it was processed. To this end, we leveraged high-resolution structures of INO80[44,46,47] and asked which remodeler elements might function as reader of genome information.

Recent structural and biochemical studies revealed an extended configuration of the INO80 multi-subunit architecture on mononucleosomes (Supplementary Fig. 1f): the INO80 core module (Ino80 protein containing the Snf2-type ATPase, Ies2, Ies6, Arp5, Rvb1, Rvb2) engages the nucleosome core particle[44,46]. The nuclear actin-containing Arp8 module (Ino80-HSA domain, Arp8, Arp4, nuclear actin, Ies4, and Taf14) binds along 40–50 bp of linker DNA at the nucleosome entry site[44,46,48], while the species-specific Nhp10 module (Nhp10, Ies1, Ies3 and Ies5) bound to the Ino80 N-terminal region is located at the distal site of INO80's linker DNA footprint[48]. Linker DNA binding by the Arp8 and Nhp10 modules was proposed to provide a DNA linker length dependent sensor that is allosterically coupled to processive nucleosome translocation catalyzed by the INO80 core[47–49]. In vivo ChIP-exo mapping suggested a highly similar INO80 configuration at +1 nucleosomes with the Arp8 or Nhp10 modules located at adjacent promoter regions[41]. Thus, we reasoned that these INO80 modules are prime candidates for reading genomic DNA sequence information.

To test this hypothesis, we targeted candidate INO80-DNA interactions based on the high-resolution cryoEM and X-ray structures of the INO80 core and Arp8 module, respectively, as well as on homology modeling of the structurally less well characterized Nhp10 module. For the INO80 core, we tested the role of ATP hydrolysis by the hetero-hexameric AAA+-ATPase Rvb1/2 (Fig. 2c and Supplementary Fig. 1c), which structurally organizes the nucleosome core binding and remodeling unit of INO80[44,46]. For the Arp8 module, we employed the Ino80-HSA helix mutants, which contain substitutions of highly conserved lysine/arginine to glutamine residues in the HSAα1 and/or HSAα2 helices (HQ1, HQ2, and combined HQ1/2 mutants, respectively) that are important for linker DNA binding[47] (Fig. 2c and Supplementary Fig. 1e). For the Nhp10 module, we either mutated site-specifically the HMG box II in Nhp10 based on well-known DNA binding activity of HMG box proteins or removed the entire Nhp10 module by omitting Nhp10, Ies1, Ies3 and Ies5

or truncating Ino80's N-terminal 1-461 residues, to which this module binds (Fig. 2c and Supplementary Fig. 1e, g, h). This latter mutant corresponded to the *Chaetomium thermophilum* INO80 core complex used in the cryoEM structure[44], which we also employed here. Nhp10 module HMGII box and Arp8 module HQ1 or HQ2 mutations were also combined (HMGII-HQ1, HMGII-HQ2 mutants, respectively) (Fig. 2c and Supplementary Fig. 1e).

**The INO80 Arp8 module is a reader of genomic sequence information.** Comparison of nucleosome patterns in aligned heat map or composite plots suggested that most INO80 mutant complexes generated similar +1 nucleosome positioning as WT INO80 (Fig. 2d, e and Supplementary Fig. 1c). Rvb1/2 ATPase activity was not required (Supplementary Fig. 1c), consistent with the likely role of Rvb1/2 during INO80 biogenesis[50]. Even the heterologous *C. thermophilum* INO80 core complex (ctINO80^ΔN) appeared to generate +1 nucleosomes on the *S. cerevisiae* genome to a remarkable extent, suggesting a conserved readout mechanism (Fig. 2d, e). Only the HQ1/2 double mutant complex was substantially impaired in +1 nucleosome positioning (Fig. 2d, e), consistent with its impaired nucleosome sliding and decoupled ATPase activity[47]. The apparent robustness of INO80's +1 nucleosome positioning activity was in contrast to effects on the nucleosome spacing and alignment activity, which was affected for most of these INO80 mutants (see accompanying paper[39]).

Quantification of distances between +1 nucleosome positions reconstituted in vitro and observed in vivo revealed a distinct impact of INO80 mutations (Fig. 2f, g). Paired-end sequencing enabled accurate determination of nucleosome dyad positions on individual DNA molecules, and we included also a lower histone-to-DNA mass ratio (~0.2, accompanying paper[39]) than mostly used in this study (~0.4) to further reduce possible next-neighbor nucleosome effects. WT INO80 and Nhp10 module mutants generated in vivo-like +1 nucleosomes with remarkable precision (Fig. 2f, g), whereas INO80 complexes bearing the HQ1 mutation and the ctINO80^ΔN complex generated +1 nucleosome positions that deviated more from the in vivo positions than those generated by the other complexes (Fig. 2f). Compared to WT INO80, +1 nucleosome positioning by complexes with the HQ1 mutation was shifted by 10 bp downstream and reduced positioning precision was reflected in broadened distributions, which suggested that DNA sequences underlying in vivo +1 nucleosome positions correspond more to the DNA sequence preferences for nucleosome positioning of the WT versus the

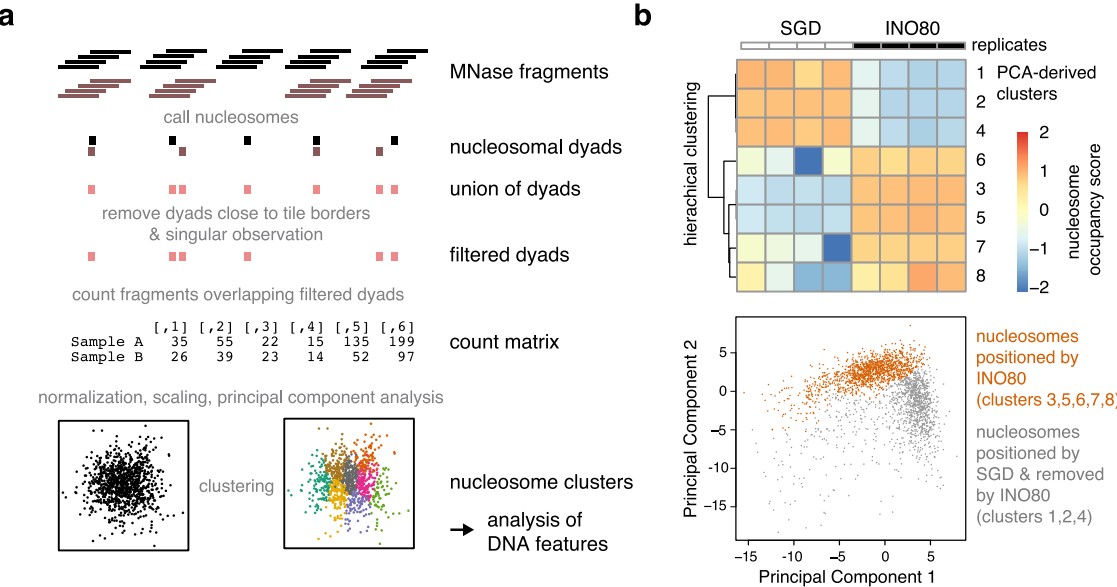

**Fig. 3 Principal component analysis (PCA)/clustering approach. a** Schematic of the analysis by using two conditions (black and gray) as an example. For details see main text and Methods section. **b** PCA/clustering approach applied to paired-end MNase-seq data of eight replicates of SGD chromatin prepared with embryonic *D. melanogaster* histones at histone-to-DNA mass ratio of 0.4 and either alone (SGD; *n* = 4) or after incubation with *S. cerevisiae* WT INO80 complex (INO80; *n* = 4). For paired-end sequencing datasets see Supplementary Data 2 and GEO deposition at GSE145093. In total, 2393 filtered nucleosome dyad positions were derived. Based on sequencing reads, nucleosome positions were subjected to a PCA from which eight clusters were derived by K-means clustering. Nucleosome occupancy scores were calculated for each replicate condition in a PCA-derived cluster. Based on these scores a row scaled heatmap (upper panel) was generated by red–white–blue color coding and hierarchically ordering in order to visualize the outcome of the PCA/clustering analysis. INO80 preferentially positioned nucleosomes at genomic positions identified in clusters 3, 5, 6, 7, and 8, which constitute together the group of nucleosomes positioned by INO80, while the respective positions were not frequently occupied in SGD chromatin. Conversely, clusters 1, 2, and 4 identified genomic positions that were frequently occupied by nucleosomes in SGD chromatin, but remodeled into different positions by INO80. The two groups of nucleosome positions are clearly distinct in a principal component 1 (PC1) versus PC2 plot as shown in the lower panel (gray: nucleosome of clusters 1, 2, and 4; orange: nucleosome of clusters 3, 5, 6, 7, and 8). Taken together, the analysis reveals that INO80 remodeling alters almost the entire landscape of respective nucleosome positions. DNA features of nucleosomes belonging to the two groups of clusters were further analyzed as presented in Fig. 4.

mutant INO80 complexes (see below) (Fig. 2f, g). Such downstream shifts, observed here for individual INO80 point mutations, were reminiscent of similar effects resulting from INO80 depletion in the context of the interplay with other remodelers in vivo[20,28,41,51].

Taken together, our mutational analysis of candidate DNA contacts indicated robust processing of genomic sequence information by INO80 with a decisive role of the Arp8, but not the Nhp10 module, as direct reader of genome information at promoters.

**DNA shape/mechanics readout underlies nucleosome positioning by INO80.** Based on our mutational analysis on the protein side of the interactions, we turned to the DNA side and sought to identify genomic DNA sequence features that provide positioning information. Previously, we proposed that *S. cerevisiae* INO80 might read DNA shape features of nucleosomal DNA[29]. However, this hypothesis was based on correlation and the approach limited further interpretation, mainly because we used gene ranking by MNase-seq signal strength at predefined +1 to +3 nucleosome regions before and after remodeling as the discriminating category. This may introduce a bias toward the starting conditions, i.e., DNA sequence preferences of histones and variations in SGD assembly conditions. Moreover, the analysis was limited to predefined regions and numerous other DNA sequence motifs present at gene starts, e.g., evolved in the context of transcription regulation, may have convoluted the search for positioning information.

Here, we overcame these limitations and searched for the DNA sequence features of nucleosome positioning preferences by

INO80 more globally, not only at promoters, and explored by a structure-based mutational analysis the direct and causal impact of altered INO80-DNA contacts on these preferences. We established a sensitive and unbiased principal component analysis (PCA)/clustering approach solely on the basis of de novo generated nucleosome dyad positions determined by paired-end sequencing. This enabled unsupervised PCA/clustering of a large number of datasets (e.g., replicates, different assembly degrees, INO80 WT, and various mutant complexes, etc.) without prior assumptions (Fig. 3a).

Nucleosomes remodeled by WT INO80 clearly clustered differently after PCA than those assembled during SGD without remodeling (Fig. 3b), i.e., this approach could clearly distinguish positioning preferences under different conditions. The DNA sequences in different clusters did not differ in terms of sequence motifs assessed by motif search algorithms like Homer and Meme (expectation value threshold 0.05), in contrast to previous studies of an isolated, truncated construct of the human INO80 HSA domain that indicated sequence-specific DNA binding[52].

However, DNA sequence information need not result in classical sequence motifs but may correspond to DNA shape features that are encoded in a more redundant way, i.e., rather disparate sequences may share similar shape features[53]. A composite plot of the DNA shape feature propeller twist of SGD-reconstituted versus INO80-remodeled nucleosomes revealed symmetrical but strikingly different profiles (Fig. 4a), revealing distinct DNA sequence requirements for INO80- and SGD-mediated positioning. Whereas propeller twist is largely affected by the number of intra-bp hydrogen bonds, other shape

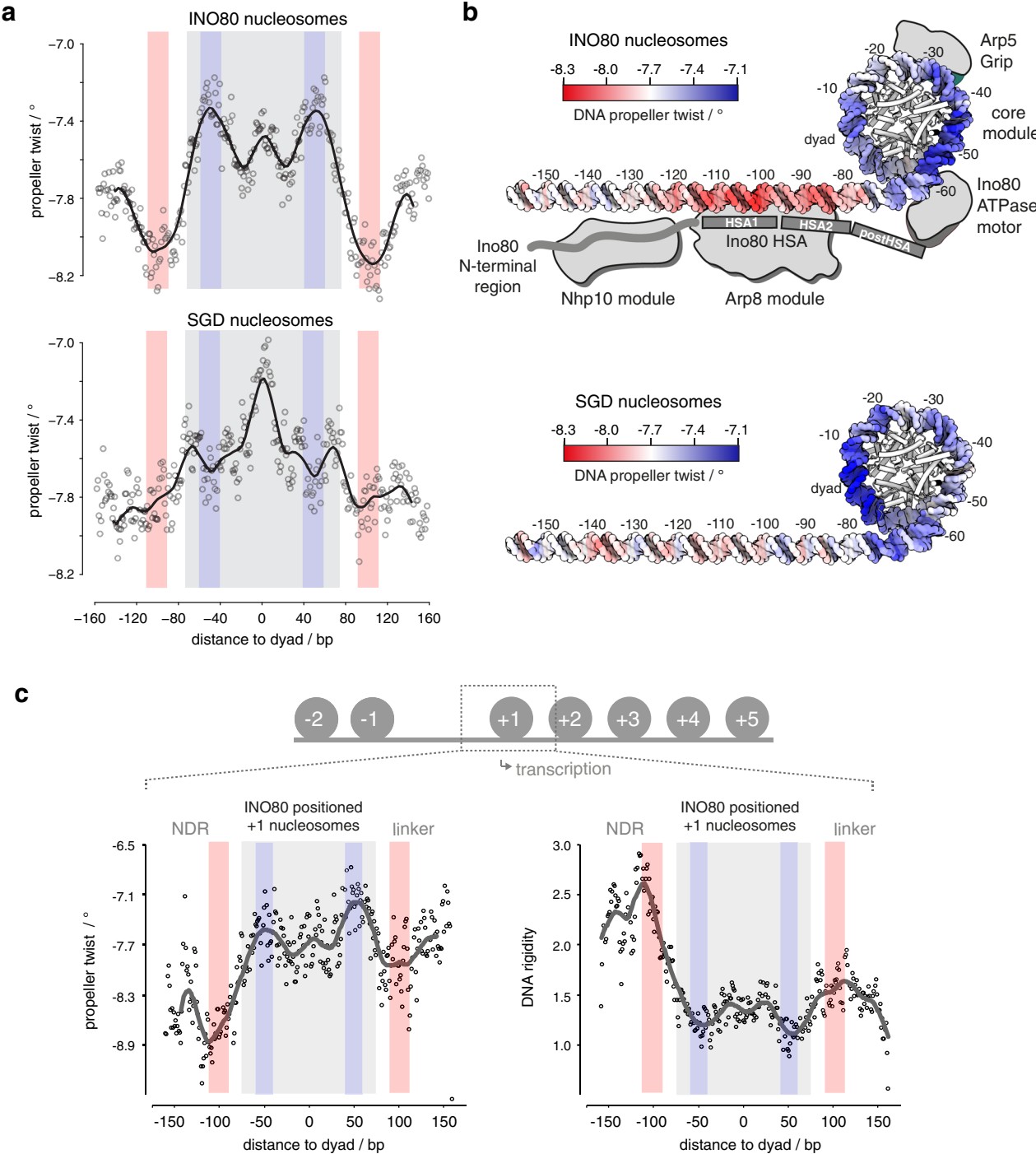

**Fig. 4 DNA shape/mechanics readout underlies nucleosome positioning by INO80 and SGD. a** Propeller twist DNA shape profiles for nucleosomal sequences occupied by histones in SGD chromatin with (INO80 nucleosomes, 1295 sequences, clusters 3, 5, 6, 7, and 8 as defined in Fig. 3b) or without (SGD nucleosomes, 1098 sequences, clusters 1, 2, and 4 as defined in Fig. 3b) remodeling by recombinant *S. cerevisiae* WT INO80 complex. Light red and light blue background indicate regions of major differences between SGD and INO80 profiles. Light gray background marks the extent of the nucleosome core particle. **b** Red–white–blue color gradient mapping of propeller twist DNA shape profile from **a** on model of linker and nucleosomal core DNA. Binding architecture of INO80 is shown schematically and based on structural data[44,47] and biochemical mapping[48]. **c** Propeller twist DNA shape and DNA rigidity profiles for INO80-positioned +1 nucleosomes, all with the same orientation relative to the direction of transcription. See main text and "Materials and methods" for a description of the DNA rigidity score. Note that the promoter NDR around −100 bp corresponds to a rigid DNA motif, while the score indicates increased flexibility around −55 bp between the ATPase motor and the Arp5 grip of INO80 (see **b**).

features gave corresponding results (Supplementary Fig. 2a). These other shape features take into account interactions either between adjacent bp (helix twist and roll) or with additional nucleotides (minor groove width). The profile symmetry validated the shape information content as no nucleosome

orientation was to be expected and symmetrical shape profiles are unlikely to occur by chance if no underlying shape feature were involved. Importantly, similar but asymmetrically distorted shape profiles were seen for nucleosomes reconstituted at positions close to in vivo +1 nucleosome positions and oriented

relative to the direction of transcription (Fig. 4c). This defined orientation led to asymmetrical profiles and showed that such pronounced DNA shape signals are also present in +1 nucleosome regions at gene promoters and strongly suggested that we identified the DNA-encoded signal for INO80-mediated +1 nucleosome positioning. The structural readout of DNA features, both in the promoter NDR as well as in the +1 nucleosome, is also consistent with in vivo binding of INO80 subunits to such regions, as observed by ChIP-exo mapping[41].

DNA shape profiles establish nucleosome positioning information that is distinct from previously known DNA sequence preferences of histones. The relevance of DNA shape for remodeler-mediated nucleosome positioning was further underscored by a striking congruency between our PCA/clustering data and prior mechanistic insights derived from high-resolution structural information and biochemical mapping. The shape profiles of SGD-reconstituted versus INO80-remodeled nucleosomes differed mostly in the ±55 bp and ±100 bp regions relative to the dyad (color shaded areas in Fig. 4a) where functionally important interactions with the INO80 complex are suggested by the independent biochemical and structural information available from INO80 in complex with mononucleosomes (Fig. 4b). The HSA helix at the Ino80 N-terminus contacts linker DNA at about −100 bp from the dyad[47,48]. The −55 bp region from the dyad lies between the Ino80 ATPase domain and the DNA contact point of Arp5. Both of these regions are critically important for nucleosome translocation. DNA strain build-up in the −55 bp region by successive rounds of DNA pumping by Ino80 ATPase motor is a central element of the proposed core mechanism of nucleosome translocation by INO80, while sensing of linker DNA by the Arp8 module ensures allosteric coupling of ATP hydrolysis to DNA translocation, which was proposed to prevent back-slippage during DNA strain build up[44,48].

This congruency immediately suggests a molecular mechanism by which an active readout not only through recognition of ground-state average DNA shape features, but also via ATP hydrolysis-driven perturbation of mechanical properties of DNA leads to the positioning of nucleosomes. The most immediate mechanical property of the double-helix is conformational flexibility. To assess this property on a genomic scale, we introduced a DNA rigidity score that characterizes how rigid/flexible DNA is within a local region at bp resolution[33]. We considered A-tracts of consecutive ApA (TpT) or ApT steps as dominant factor in increasing rigidity due to strong stacking interactions combined with inter-bp hydrogen bonds in the major groove[32,54]. The rigidity score accounts for the length of A-tracts as longer runs of ApA (TpT) and ApT steps without TpA steps or G/C bp increase rigidity of a DNA fragment. We observed that DNA rigidity is correlated with DNA shape features, and the correlation remains at a consistent level across all positioned nucleosomes (Supplementary Fig. 2a–c). This analysis reveals that +1 nucleosome positioning by INO80 involves placement of nucleosomes where DNA flexibility is increased at the −55 bp region between the ATPase motor and the Arp5 grip, while the promoter NDR region harbors a rigid DNA element where the Arp8 module is located (Fig. 4c). Intriguingly, a similarly rigid promoter DNA motif at the same distance in respect to the +1 nucleosome was also identified in a parallel study, where DNA mechanics were measured experimentally on a genomic scale via library-based DNA circularization assays[55].

**Altered Ino80-HSA-helix-DNA contacts affect DNA shape/mechanics readout by INO80.** To establish causality, we probed whether the INO80-DNA contacts and different histones would affect the readout of DNA shape/mechanics. Nucleosomes positioned by WT INO80 clustered together with those positioned by mutant complexes where mutations affected the Nhp10 module, i.e., the Ino80 N-terminus or Nhp10 module subunits including the Nhp10 HMG Box (Fig. 5a). This corroborated our results regarding nucleosome positioning in promoter regions (Fig. 2d–f) and ruled out a major role for the Nhp10 HMG box in DNA shape/mechanics readout by INO80. In contrast, all mutant complexes impaired in HSA helix-DNA contacts, either the HQ1 or HQ2 mutation and each also in combination with the HMGII mutations, generated distinct clusters of nucleosome positions (Fig. 5a). Overall shape/mechanics preferences were not much affected if endogenous fly versus recombinant human histones were used (Fig. 5b). This validated our use of fly histones for the comparisons among WT and mutant INO80 complexes in this approach.

In total, there were three major classes of nucleosome positions, those generated by (1) SGD, (2) WT INO80/Nhp10 module mutant complexes, or (3) HSA helix mutant complexes (Fig. 5a). To investigate the differences in DNA sequence preferences only between the INO80 complexes and at minimal contribution of neighboring nucleosomes, we clustered only the respective samples with low assembly degree SGD chromatin (Fig. 5c) and compared the resulting DNA shape/mechanics profiles of clusters with clearly different occupancies among the INO80 complexes, e.g., cluster 1 versus 3 (Fig. 5c and see Supplementary Fig. 3 for all clusters). Propeller twist signal profiles clearly differed between clusters that contained nucleosome positions preferentially generated by the HSA helix-mutated INO80 versus WT or Nhp10 module mutated complexes. In particular, the ±100 bp region of the linker DNA showed a distinct shift of the propeller twist signal by more than 20 bp between clusters 1 and 3 (Fig. 5d). As this is the region where the Ino80 HSA domain contacts DNA (Fig. 4b), these data directly showed that these HSA helix-DNA contacts contributed to the DNA shape/mechanics readout during nucleosome positioning. Moreover, additional changes of propeller twist signals within the nucleosomal DNA region provided, in context of Ino80 HSA mutations, evidence for the allosteric interplay between the Arp8- and the core module of INO80[47,48]. We conclude that INO80 positions nucleosomes via a readout of DNA shape/mechanics profiles. This information and its readout are distinct from known DNA sequence preferences of histones suggesting that remodelers play an active role in translating genomic information into nucleosome positions, i.e., determine nucleosome positions through DNA sequence information readout during their specific molecular mechanism of remodeling.

**The DNA sequence-specific barrier Reb1 regulates nucleosome positioning by INO80.** Having established that INO80 reads DNA shape/mechanics features and translates this information via specific modules into nucleosome positions, we asked next whether INO80 also processes nucleosome positioning information from DNA sequence-specific barriers (Fig. 1b). Reb1 is a GRF important for promoter nucleosome organization in vivo[26]. Sequence-specific GRFs serve, via an unknown mechanism, as nucleosome positioning alignment point for remodelers like ISW1a or ISW2[29]. Moreover, in vivo mapping of INO80 subunits by ChIP-exo[41] indicated that INO80 adopts an extended conformation, which may bridge Reb1 binding sites and +1 nucleosomes.

To directly address whether Reb1 binding at cognate promoter sites controls +1 nucleosome positioning by INO80, we titrated increasing Reb1 concentrations to our whole-genome reconstitutions. Reb1 binding in the nanomolar affinity range clearly

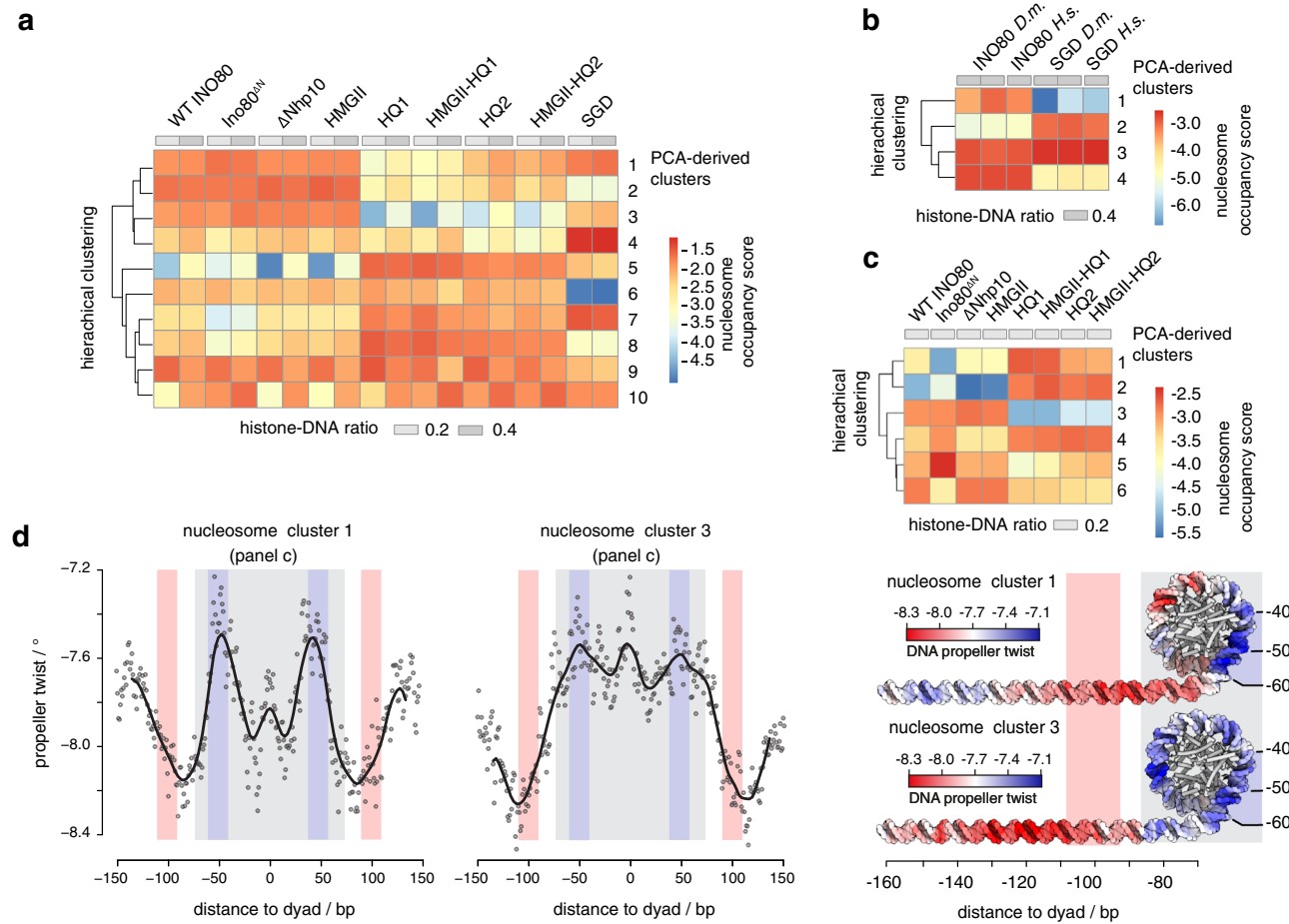

**Fig. 5 Structure-based mutations probe the DNA shape/mechanics readout by INO80. a** Heat map representation of PCA/clustering analysis as in Fig. 3b, but not row scaled and for SGD chromatin with embryonic *D. melanogaster* histones at the indicated histone-to-DNA mass ratios either without (SGD) or after remodeling by the indicated recombinant *S. cerevisiae* WT and mutant INO80 complexes (mutants defined in Fig. 2c; for paired-end sequencing datasets see Supplementary Data 2 and GEO deposition at GSE145093.) **b** As **a** but for SGD chromatin with embryonic *D. melanogaster* (*D.m.*) versus recombinant *H. sapiens* (*H.s.*) histones and only remodeling by recombinant *S. cerevisiae* WT INO80 complex (INO80). **c** As **a** but only for the indicated subset of samples. **d** Left: as Fig. 4a but only for nucleosomes from the indicated clusters of **c**. Right: as in Fig. 4b, propeller twist DNA shape data were mapped onto model of linker and nucleosomal DNA by using red–white–blue color gradient. See Supplementary Fig. 3 for all clusters.

improved nucleosome positioning by INO80 at promoters with Reb1 sites in terms of +1 nucleosome occupancy (peak height), array extent, and NDR depth (Fig. 6a, b and Supplementary Fig. 4a), again independent of histone species origin (Supplementary Fig. 4b). Detailed quantification of nucleosome spacing and array phasing at Reb1 sites and at different nucleosome densities was studied in the accompanying paper[39].

To probe the underlying mechanism by which INO80 relays positioning information from Reb1 to +1 nucleosomes, we turned to classical mononucleosome assays. We generated mononucleosomes with a long linker DNA on one side of a promoter (of gene locus yGL167c) that was selected based on INO80 and Reb1 occupancy measured by ChIP-exo in vivo[41] and clearly in vivo-like nucleosome positioning in whole-genome reconstitutions[29]. In vivo, the Reb1 site of the yGL167c promoter is 145 bp upstream of the +1 nucleosome dyad (about 72 bp to the 5′ flank of the nucleosome core particle as the distance of this flank to the dyad is about 73 bp), which matches closely the median distance of 150 ± 18 bp measured for all Reb1 promoter sites (median distance to the 5′ flank of 77 ± 18 bp, Fig. 6f; see also accompanying paper[39]). We replaced the +1 nucleosome sequence by a Widom 601-nucleosome positioning sequence (NPS)[56] and reconstituted with this construct (Fig. 6c, left) via SGD the in vivo promoter nucleosome architecture.

Reb1 was added substoichiometrically to reconstituted yGL167c-NCP601 mononucleosomes. As separation in native polyacrylamide gel electrophoresis could distinguish mononucleosomes with and without bound Reb1, we could compare remodeling kinetics with and without Reb1 in the same reaction (Fig. 6c, right). Kinetics of sliding the initially end-positioned nucleosome to the center were much slower, if at all detectable, in the presence of Reb1 (Fig. 6c, d). As the distance between bound Reb1 and the 601-nucleosome was as in vivo and therefore probably corresponded to the steady state distance set by INO80, we prepared and assayed in the same way a second construct (yGL167c-20-NCP601, Fig. 6c) with additional 20 bp of DNA inserted in the yGL167c promoter. This end-positioned 601-nucleosome was clearly moved toward the Reb1 barrier by INO80 (Fig. 6c), but again at a slower rate compared to sliding this nucleosome to the center in the absence of Reb1 (Fig. 6d).

We asked next, whether decreased sliding kinetics were caused by inhibition or by decoupling of ATPase activity. Notably, most INO80 mutations that abrogated nucleosome sliding, such as the HQ1/2 or Arp5 mutations, still showed robust ATPase activity[44,47]. In contrast, INO80 ATPase assays in the presence of yGL167c-NCP601 mononucleosomes showed about twofold decreased ATPase activity upon addition of Reb1 compared to reactions without Reb1 (Fig. 6e). This was not a general effect of

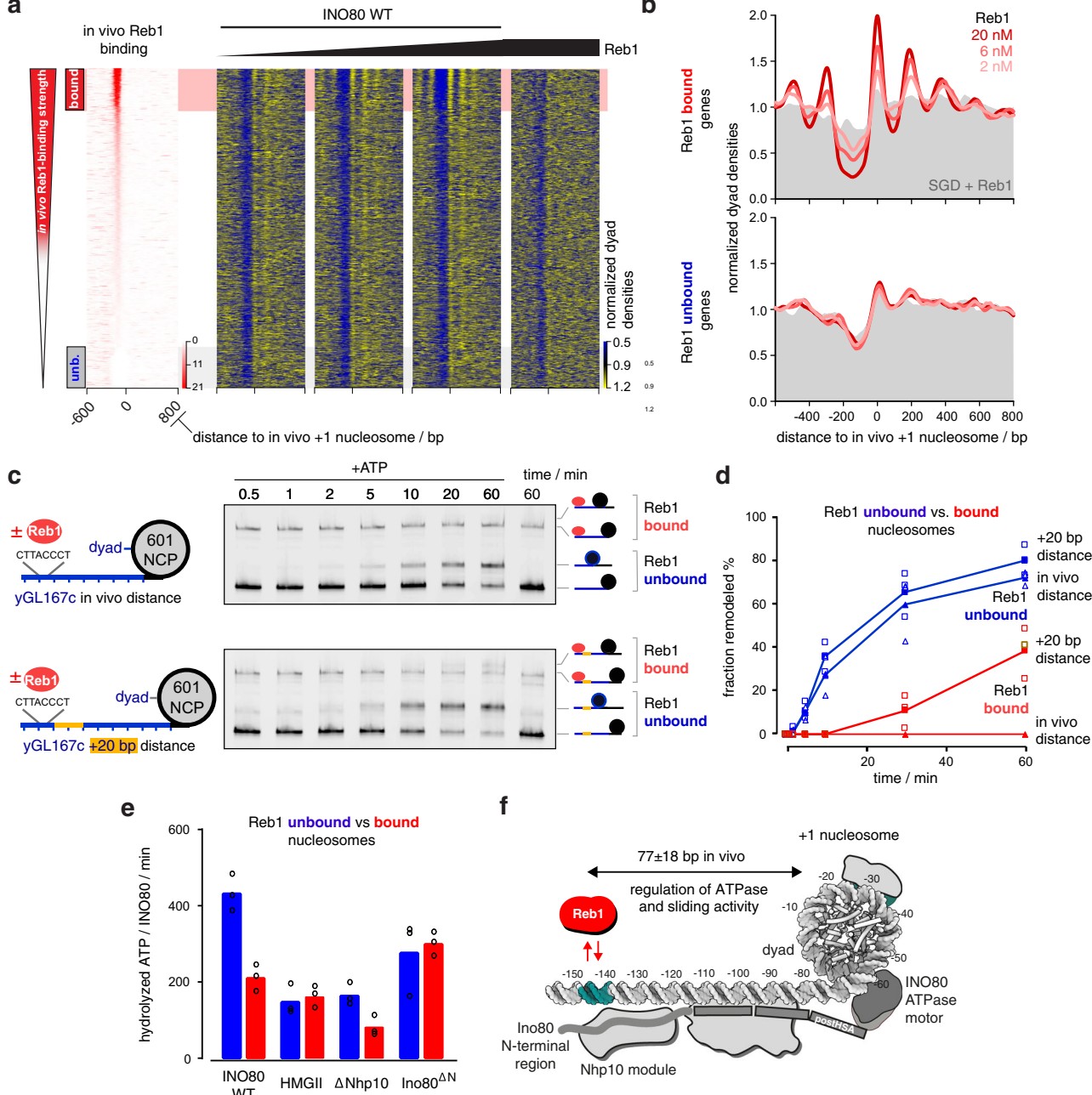

Reb1 in this assay as the HMGII as well as the Ino80$^{\Delta N}$ INO80 mutant complexes did not show a Reb1-dependent reduction of ATPase activity upon Reb1 addition (Fig. 6e). The ATPase activity of the ΔNhp10 INO80 mutant complex was still regulated by Reb1, although at lower absolute levels relative to WT INO80, which were observed for all these mutant complexes. The detailed mechanism of this intriguing allosteric communication across a distance of more than 70 bp linker DNA awaits further structural studies. However, based on the regulatory role of the Ino80 N-terminal region even in the absence of the Nhp10 module, we cautiously speculate that it might serve not only as a binding platform for Nhp10, but that it stimulates the activity of INO80 in absence of Reb1 possibly via restricting the dynamics of the Arp8 module.

Taken together, we concluded that Reb1 binding to its promoter sites regulates INO80 activity allosterically by inhibition through interaction via the N-terminal region of Ino80 that is

modulated by the Nhp10 module subunits. The multi-subunit architecture of INO80 relays thereby positioning information between Reb1 and +1 nucleosomes, adjusts the +1 nucleosome to its in vivo-like position and programs thereby genic regions for formation of nucleosome arrays (Fig. 6f).

**INO80 integrates synergistic information from DNA shape/ mechanics and Reb1 at promoters.** Next, we addressed if and how INO80 integrates different information input, from DNA shape/mechanics versus from bound Reb1, into nucleosome positions.

First, we asked if promoters with Reb1 sites solely rely on +1 nucleosome positioning information via Reb1 or also contain DNA shape/mechanics information guiding +1 positioning by INO80. In vitro INO80 positioning activity without Reb1 was compared between genes with Reb1 sites and genes without GRF sites (Fig. 7a). Although the latter may rely more on DNA shape/

**Fig. 6 Reb1 regulates nucleosome positioning by INO80 and INO80's ATPase and sliding activity. a** Heat maps of MNase-seq data for SGD chromatin assembled with recombinant *H. sapiens* histones at histone-to-DNA mass ratio 0.4, incubated with recombinant *S. cerevisiae* WT INO80 and increasing concentrations (ramp denotes 2, 6, and 20 nM) of recombinant Reb1. Right most heat map shows sample prepared with embryonic *D. melanogaster* histones. Heat maps are aligned at in vivo +1 nucleosome positions and sorted according to decreasing (top to bottom) anti-Reb1 SLIM-ChIP score (in vivo Reb1 binding[82]) shown in leftmost heat map. Horizontal red or gray shading highlights genes with strong (top 12.5%, "bound") or weak (bottom 12.5%, "unb." for unbound) in vivo Reb1 promotor-binding, respectively. Single replicates (replicate 1) were plotted, see Supplementary Fig. 4a and Supplementary Data 1 for all replicates. **b** Composite plots of MNase-seq data as in **a** averaged over 620 genes highlighted in red (Reb1 bound, top) or gray (Reb1 unbound, bottom) in **a**. Gray backgrounds show respective composite plots of SGD chromatin to which Reb1, but not INO80 WT, were added. **c** Left: mononucleosome substrate design with 80 bp (yGL167c-NCP601, top) or 100 bp (yGL167c-20-NCP601, bottom) linker DNA taken from a promoter (yGL167c) with in vivo-like +1 nucleosome positioning by INO80[29] in vitro and strong INO80 binding in vivo[41]. Guided by its dyad position, we replaced the genomic +1 nucleosome sequence of yGL167c with a 601-nucleosome positioning sequence. Right: representative example of nativePAGE nucleosome sliding assay for mononucleosome species as shown to the right of the gel image and with 90 nM mononucleosome, 45 nM Reb1, 10 nM recombinant *S. cerevisiae* WT INO80 and 1 mM ATP (denoted as +ATP). 60 min time point was also performed without the addition of ATP. Experiments were independently replicated (n = 3). **d** Quantification of sliding assays with bound (red) or unbound (blue) Reb1 from **c** and two other replicates (see Source data). Individual datapoints for independent replicates (n = 3) are shown as empty, average values as filled symbols. **e** NADH-based ATPase assay (25 nM mononucleosomes, 10 nM recombinant *S. cerevisiae* WT or mutant INO80 complex, and with (red) or without (blue) 25 nM Reb1). Individual datapoints for biological replicates (n = 3) are shown as circles; bar heights correspond to respective average values. **f** Structural data[44,47] and biochemical mapping[48] suggest a putative binding architecture of INO80 which may bridge Reb1 and +1 nucleosomes. Allosteric communication occurs across a distance of more than 70 bp (in vivo median distance of 77 ± 18 bp measured between ChIP-exo mapped Reb1 binding motifs[57] and MNase-seq derived +1 nucleosome dyads[87]; see also accompanying paper[39]).

mechanics information than the former, INO80 alone similarly generated in vivo-like +1 nucleosome positions at both promoter groups demonstrating that both must contain the respective DNA shape/mechanics information.

Second, we asked if, at the same promoter, the information from DNA shape/mechanics versus from Reb1 synergized or competed in +1 nucleosome positioning by INO80. To this end, we compared the positions of +1 nucleosomes at Reb1 site-containing promotors in the absence versus presence of Reb1. Upon addition of Reb1, we observed, regardless of histone species origin, not only clearly improved +1 nucleosome positioning in terms of MNase-seq peak heights, but also very similar +1 positions (Fig. 7b) arguing for a synergy between both types of positioning information. We probed this synergy at promoters further by utilizing different nucleosome densities in our reconstitutions, as we found that the distance between INO80 and Reb1 at all genomic Reb1 sites varied with nucleosome density (see accompanying paper[39]). In addition to medium nucleosome density (histone-to-DNA mass ratio 0.4, Fig. 7b) we sampled also low and high nucleosome density (histone-to-DNA mass ratio 0.2 and 0.8, respectively) and quantified how close composite +1 nucleosome peak positions set by INO80 in the presence or absence of Reb1 were relative to the in vivo +1 positions (Fig. 7c). DNA shape/mechanics-guided positioning by INO80 in absence of Reb1 robustly generated in vivo-like +1 positions at all densities. Reb1-guided +1 positions at low and medium nucleosome density were similarly in vivo-like as they differed on average by 6 ± 3 bp from the DNA shape/mechanics-guided positions, which was within the experimental error of our reconstitutions. However, at high density Reb1-guided positions were closer to Reb1 than in vivo and differed on average by 15 ± 5 bp from the +1 positions reconstituted without Reb1.

Together, we concluded that genome sequence must have evolved DNA shape/mechanics information downstream of a Reb1 site in direction of transcription so that +1 nucleosome positioning by INO80 alone is hard-wired very close to the average in vivo +1 position and that this DNA shape/mechanics information may synergize with guiding by Reb1 at low or medium nucleosome densities. At higher nucleosome densities, this synergy was compromised either due to dominant positioning signal from Reb1 or a weaker signal from DNA shape/mechanics. In the accompanying study[39], we show, in the context of the remodeler ruler concept, that the latter was the case. As promoter Reb1 sites are situated in vivo within NDRs[57], which,

by definition, represent regions of locally lower than average nucleosome density, the synergy between Reb1- and DNA shape/mechanics-guided +1 positioning may be a common default state at many promoters. Nonetheless, our data demonstrate that regulation of nucleosome density may allow tuning of Reb1-aligned +1 nucleosome positioning (see "Discussion").

Interestingly, the synergy between DNA shape/mechanics- and Reb1-guided nucleosome positioning by INO80 at medium nucleosome density only affected the +1 but not the −1 nucleosome position, i.e., there was a shift in −1 positions upon Reb1 addition (Fig. 7b). In addition, relative nucleosome occupancy in terms of MNase-seq peak heights generally increased around the Reb1 sites in the presence of Reb1 (Fig. 7b), but relatively more for the +1 than the −1 nucleosome. Therefore, we hypothesized that this asymmetrical effect of bound Reb1 on the +1 versus −1 nucleosomes was linked to the direction of transcription and maybe also to the direction of the non-palindromic Reb1 PWM. To directly address this, we grouped Reb1 site-containing promoters by the relative orientation of Reb1 PWM and TSSs toward each other and compared INO80-generated nucleosome patterns in the absence or presence of Reb1 (groups 1–3, Fig. 7d). Peak heights were generally higher in the presence than in the absence of Reb1 and peak positions were always symmetrical around Reb1 in these groups as quantified in our accompanying paper[39]. Nonetheless, the peak heights of the −1 position increased relatively less than that of the +1 nucleosome, and the −1 position shifted in plus versus minus Reb1conditions, as seen in Fig. 7b. This asymmetry correlated with the orientation of the TSS and not of the Reb1 PWM motif and was not seen for bidirectional promoters (group 3). This provided evidence for our conclusion that synergistic DNA shape/mechanics information evolved next to Reb1 sites only in places where a +1 nucleosome becomes positioned that plays the well-known role in regulation of transcription initiation[4,28]. Indeed, our analysis revealed that promoters in groups 1 and 2 exhibited asymmetrical DNA shape/mechanics features and strand-specific poly(dA)/(dT) prevalence in the direction of transcription (Fig. 7e), while this was not the case for bidirectional promoters (group 3).

Overall, we concluded that INO80-mediated nucleosome positioning was symmetrically guided on either side of Reb1 regardless of the Reb1 PWM orientation. However, in terms of nucleosome positioning and occupancy, there was an enhancing effect at low and medium nucleosome densities if DNA shape/

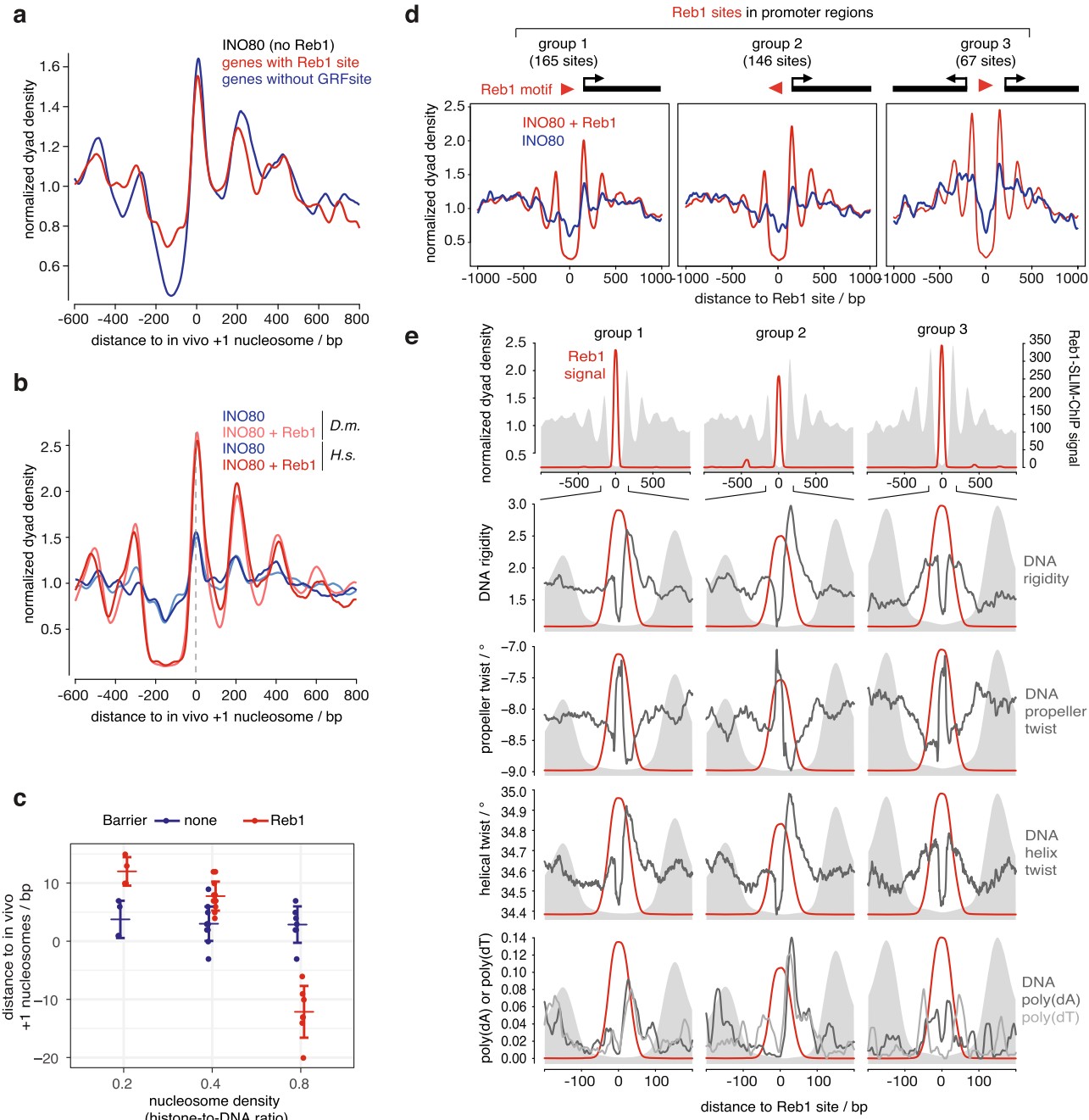

mechanics features synergized in guiding to the same nucleosome position flanking the Reb1 site. This synergism evolved next to Reb1 sites only in direction of transcription, and INO80 integrates the input from both kinds of information, while it generates in vivo-like +1/−1 nucleosome positioning patterns.

**DNA ends are potent barriers for INO80 nucleosome positioning.** Having established a synergy between DNA shape/mechanics and Reb1 sites at gene promoter regions, we asked whether we can uncouple barrier-mediated positioning from a promoter sequence context. To test this idea, we analyzed nucleosome positioning at all in vivo mapped genomic Reb1 sites (Fig. 8a, b). Consistent with our findings above, we observed symmetrical nucleosome arrays around all Reb1 sites (Fig. 8b, top right) suggesting that barrier-mediated positioning can occur independently of other DNA sequence features. In light of this,

we considered that INO80 may align nucleosomes also to different barrier types as long as they represented a clear alignment point. In our search of the minimalistic system that provides nucleosome positioning information, we wondered if simply a DNA end could constitute a barrier. Notably, INO80 has been involved in DNA damage response signaling upon DNA double strand breaks (DSBs) in vivo[58]. In principle, such as scenario was already tested in classical mononucleosome sliding assays as these automatically involve two DNA ends. However, effects there may have been due to the comparatively short length of template DNA and to the presence of two DNA ends in close vicinity. Our genome-wide system allowed us to test the effect of one-sided DNA ends in the context of very long DNA. We introduced double stranded DNA ends at fortuitous locations, i.e., without likely evolutionarily shaped context, throughout the *S. cerevisiae* genome via restriction enzyme (RE) digest of the plasmid library

**Fig. 7 INO80 synergistically integrates nucleosome positioning information from DNA shape/mechanics and Reb1 barriers. a** Composite plots as in Fig. 6b, but merge of four replicates (Supplementary Data 1) of SGD chromatin with recombinant human histones at histone-to-DNA mass ratio 0.4 incubated with recombinant *S. cerevisiae* WT INO80 plotted for either 456 genes with promoter Reb1 PWM sites or for a randomly selected but same number of genes with no GRF-PWM sites (Reb1, Abf1, Rap1, Mcm1, Cbf1[88]) in their promoters. **b** As in **a** but for matched replicate (replicate 8, Supplementary Data 1) comparing SGD chromatin with embryonic fly (*D. m.*) or recombinant human (*H. s.*) histones, ± 20 nM Reb1 and for 620 genes with promoter anti-Reb1-SLIM-ChIP sites (as in Fig. 6a, red shading, and in Fig. 6b, top). **c** Distributions of distances between +1 nucleosome positions at 620 Reb1 site-containing promoters in vivo and reconstituted by incubation of SGD chromatin with the indicated histone-to-DNA mass ratios with recombinant *S. cerevisiae* WT INO80 in the presence (Reb1) or absence (none) of 20 nM Reb1. Dots show independent replicates ($n = 4$ or 12 or 7 for histone-to-DNA ratio of 0.2 or 0.4 or 0.8, respectively, and for ±Reb1 each; either recombinant human or endogenous fly embryo histones were used, see Supplementary Data 1 and GEO deposition at GSE140614. For histone-to-DNA ratio 0.2 and 0.8 data and for detailed description and more examples of such distances between nucleosomes and barriers and their dependencies on nucleosome density and remodelers see accompanying paper[39]. Larger horizontal bars represent mean, error bars standard deviation. **d** As in **b** but only SGD chromatin with recombinant human histones incubated with recombinant *S. cerevisiae* WT INO80 ± 20 nM Reb1.MNase-seq data were merged (four replicates without and seven replicates with Reb1 (Supplementary Data 1)) and aligned at Reb1 PWM sites of groups, which were defined according to promoter uni- versus bidirectionality and Reb1 PWM orientation as indicated. Only Reb1 sites in promoters and only if identified by both anti-Reb1-SLIM-ChIP and Reb1 PWM were used (see also accompanying paper[39]). **e** Reb1 PWM-aligned composite plots for gene groups as in **d**. From top to bottom: anti-Reb1-SLIM-ChIP signal (Reb1 signal) and zoom in on distribution of DNA rigidity, propeller twist and helix twist DNA shape features, and positions of poly(dA) or poly(dT) elements (hexa-homopolymeric stretches) around Reb1 sites, each with Reb1 signal. Gray background in all panels shows composite plot of MNase-seq data with Reb1 as in **d**.

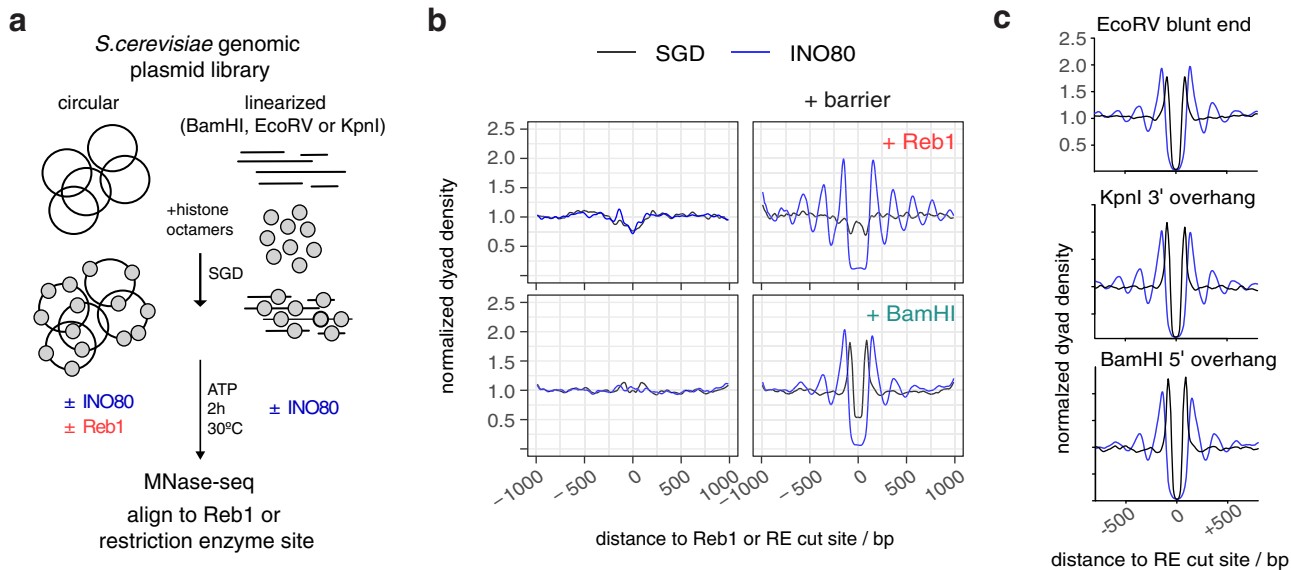

**Fig. 8 DNA ends are potent barriers for nucleosome positioning by INO80. a** Overview (analogous to Fig. 2a) of reconstitution with circular versus RE-precleaved plasmid libraries. **b** Composite plots (merged replicates, $n = 2$) of MNase-seq data aligned at barrier sites, i.e., anti-Reb1-SLIM-ChIP-defined Reb1 sites or BamHI sites for: top, SGD prepared with circular plasmid library and incubated without (SGD) or with recombinant *S. cerevisiae* WT INO80 (INO80) and with (+Reb1) or without 20 nM Reb1 as indicated, and bottom: as top but with BamHI-precleaved library if indicated (+BamHI). **c** As **b**, but for SGD chromatin with plasmid libraries precleaved with the indicated RE and data aligned at the indicated RE cut sites and only one replicate each (Supplementary Data 1). Strong peaks flanking cut RE sites in SGD chromatin without INO80 remodeling reflected an MNase-seq bias, i.e., due to the pre-cleavage, the probability is increased that MNase releases a mononucleosomal fragment with the cut site as one end relative to releasing fragments from other genomic regions.

prior to SGD reconstitution (Fig. 8a). As expected, SGD chromatin neither with nor without remodeling by INO80 showed distinct nucleosome patterns at uncleaved BamHI sites (Fig. 8b, bottom left). However, strong and symmetrical arrays were aligned at cut sites by INO80 (Fig. 8b, bottom right). The same was true for other REs that generated different kinds of DNA ends (Fig. 8c). We concluded that all three kinds of DNA ends (blunt, 3′ or 5′ overhang) were strong nucleosome positioning barriers for INO80.

## Discussion

In this study, we identified and probed the fundamental molecular determinants by which ATP-dependent chromatin remodelers position nucleosomes across the genome. An integrated approach combining fully recombinant, de novo whole-genome

reconstitutions, high-resolution structural information, and PCA/clustering analysis revealed that the INO80 complex processes DNA sequence information, both via readout of a distinct DNA shape/mechanics signature motif, as well as via alignment against a DNA sequence-specific barrier factor like Reb1 or at DSBs. INO80's multi-subunit architecture integrates the readout of different positioning information, contributes through its mechanism its own information, and determines thereby how this is translated into positions of +1 and other nucleosomes (Fig. 9).

Although the pivotal role of remodelers in chromatin organization and their dependency on DNA sequences has been recognized[29,31,59], NPSs were usually defined as sequences of intrinsic positioning by SGD driven solely by histone octamer-DNA interactions, as illustrated by the Widom 601 NPS[56]. PCA/clustering analysis enabled us now to reassess these classical SGD-

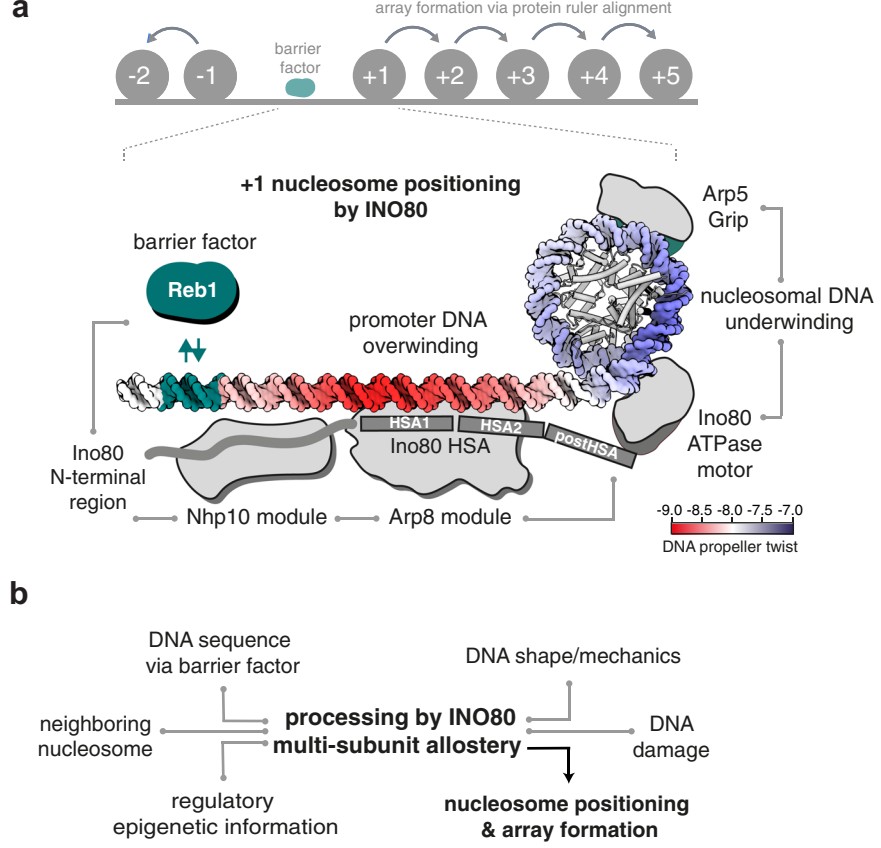

**Fig. 9 Model of +1 nucleosome positioning by INO80. a** INO80 synergistically processes genomic information derived from DNA shape/mechanics as well as DNA sequence motifs bound by GRFs, like Reb1, into +1 nucleosome positioning. Structural[44,47], biochemical[48], and ChIP-exo mapping[41] data suggest a binding architecture of INO80 at +1 nucleosomes that is fully consistent with the identified positioning information and mechanism. Promoter DNA overwinding and nucleosomal DNA underwinding is derived from the direction of DNA translocation by the Snf2-type ATPase of INO80[43]. Allosteric communication is indicated by gray lines. **b** Signal readout, integration, and processing by multi-subunit allostery within INO80 lead to nucleosome positioning and array formation. Epigenetic information such as histone marks are expected to provide an additional layer of regulatory input, e.g., in response to the physiological state of the cell.

NPSs. We find that SGD-NPSs correspond to distinct DNA sequence-dependent shape/mechanics profiles, while nucleosome positioning by a remodeler like INO80 corresponds to a different shape/mechanics profile. Therefore, we identified the latter as INO80-NPSs.

Respective remodeler-NPSs are likely to exist for other remodelers and it will be interesting where they evolved in genomes. The mere observation that INO80 and RSC remodelers generate different nucleosome positions, despite working on the same histone octamers and DNA sequences, suggested previously[29,60] that remodelers do not just allow histone octamers to occupy their thermodynamically preferred positions (otherwise different remodelers would generate the same positions), but that remodelers, as demonstrated in this study, read genomic information, actively override octamer preferences and shape the positioning landscape in a remodeler-specific way. In analogy to the genomic code mechanism for nucleosome positioning, i.e., the proposed evolution of SGD-NPSs (Fig. 1a), evolved remodeler-NPSs would implement a remodeler code mechanism for nucleosome positioning as proposed earlier[60]. We abstain from adding another code to the troubled epigenetics discussions but point out the conceptual analogy.

Importantly, we go here beyond a mere correlation between INO80-NPSs and DNA shape/mechanics profiles. The causal mechanistic link was directly established as two orthogonal

approaches converged in identifying the same critical INO80-DNA contacts and by tuning the INO80 DNA shape/mechanics readout via corresponding targeted INO80 mutations. High-resolution structures coupled with remodeling assays[44,47] suggested the Ino80-HSA-DNA contacts at −100 bp from the nucleosome dyad and the −55 bp region between the Ino80 core ATPase and the Arp5 grip as critical for the INO80 remodeling mechanism. Here, we found independently by unbiased PCA/clustering analysis that these same regions differed most in their DNA shape/mechanics features between SGD-NPSs and INO80-NPSs. Further, mutation of Ino80-HSA-DNA contacts at −100 bp caused altered nucleosome positioning patterns and altered DNA shape/mechanics features right in the same region and at the −55 bp region. Together, our results provide strong evidence for a readout of these DNA shape/mechanics features by the Ino80-HSA helix/Arp8 module and suggest a critical role of DNA shape/mechanics in regulating the build-up of DNA strain in the −55 bp region during the core mechanism of nucleosome translocation[44,48,49]. The effects at both regions are coupled via two allosteric communication pathways of possibly equal importance: on the protein side, linker DNA recognition by the Arp8 module is coupled to the activity of the Ino80 ATPase motor of the core module via the extended helical configuration of the HSA and postHSA domains[47]. On the DNA side, DNA shape/mechanics features at the histone-bound −55 bp region are

most likely coupled to DNA shape/mechanics features at the DNA linker −100 bp region in the context of DNA over- and underwinding in front and behind the Ino80 ATPase motor[38,44]. More generally, our data illustrates a regulatory circuitry comprising a two-way relationship between a molecular machine working on DNA and DNA properties feeding back into the regulation of the molecular machine. INO80-NPSs represent the nucleosome positioning information that emerges from the combination of DNA, histones, and the active interpretation via the allosteric communication within the remodeler.

For these reasons, the DNA shape/mechanics readout by INO80 importantly expands the scope of recently discussed DNA shape contributions. DNA shape was mostly studied in the context of static DNA binding, e.g., by transcription factors and GRFs[61–63]. In contrast, INO80 dynamically reads and interprets DNA shape/mechanics while tracking along DNA in an ATP-dependent manner. INO80 actively probes the mechanical properties of DNA. This read out of genome information is expected to serve as a role model for other factors that translocate along DNA or also RNA, like other remodelers, helicases, cohesins, or polymerases. For example, RNA polymerase I was suggested to read the DNA bend at its promoters[64] and RNA polymerase II may recognize its promoters via structural DNA features (bending, meltability, flexibility) rather than via classical consensus sequences[65]. As these structural properties are redundantly linked to DNA sequence, we propose that readout of such DNA structural properties may be common if factors deal with a wide range of genomic regions.

As alternative DNA sequence signals, there is DNA sequence information of classical consensus motifs for specific binding by cognate factors. GRFs are well-known to program +1 nucleosome positioning and formation of genic nucleosome arrays in vivo[26,34,66]. In light of our finding that DNA ends are also potent nucleosome positioning barriers, it is tempting to speculate that remodelers involved in DNA damage response, such as INO80[58], may generate regular nucleosome arrays as a licensing platform at DSBs in vivo.

The mechanism by which remodelers generate arrays at barriers, i.e., read positioning information via an alignment mechanism, remained largely unknown. This study reveals that nucleosome positioning by INO80 is allosterically regulated by Reb1 at promoter sites through an interaction with the N-terminal region of Ino80 (Fig. 9a). Reb1 decreased not only nucleosome sliding, but also inhibited ATPase activity of INO80, even at a distance of −145 bp between the Reb1 site and the dyad of the +1 nucleosome. In contrast, DNA linker length sensing by INO80 at DNA ends uncouples a decrease in mononucleosome sliding from its robust stimulation of ATPase activity[47,49]. Consequently, GRFs might represent a different kind of regulatory barrier compared to DSBs, at least in the absence of the DNA repair machinery. In the accompanying study[39], we identify the Arp8 module and the Nhp10 module as a multi-layered ruler element, which measures and sets nucleosome arrays differently in respect to Reb1 sites, DNA ends, and neighboring nucleosomes. Taken together, our findings lead to a model how regulation of nucleosome sliding direction bias upon interaction with a barrier can lead to stable nucleosome positioning and array formation. The multi-subunit architecture of INO80 functions similarly to a relay: INO80 receives input via its Arp8 and Nhp10 modules and communicates this information allosterically toward the ATPase of the INO80 core, where it is translated into a nucleosome position (Fig. 9b).

The exact +1 nucleosome position impacts transcription regulation, e.g., it differs between repressed and activated promoters and influences TSS selection[4,11,28,67]. In this study, we show that these positions are robustly encoded in the genome in two ways,

i.e., both by DNA shape/mechanics features and corresponding distances to the Reb1 site. Nucleosome positioning next to Reb1 did not require DNA shape/mechanics features as it also worked symmetrically on the other side even if there was no evolved promoter. Importantly, however, in context of promoter regions, we identify a co-evolved synergy between DNA shape/mechanics signatures and Reb1 binding sites, leading to asymmetric +1/−1 nucleosome positioning, as measured by MNase-seq peak heights. This synergy provides not only robustness, but also an inroad to regulation. For example, we show that Reb1-mediated +1 nucleosome positioning is altered in response to nucleosome densities. We propose that regulation of nucleosome density at promoters, e.g., via the local activity of RSC, the major nucleosome-evicting remodeler in yeast[23], may result in regulation of +1 nucleosome positions. With high RSC activity, local promoter nucleosome density is low and +1 nucleosome positioning by INO80 coincides for DNA shape/mechanics- and Reb1-information input. Upon low RSC activity, nucleosome density is high, INO80 disregards the shape/mechanics signal and places the +1 nucleosome closer to Reb1, which corresponds to the more upstream +1 nucleosome position implicated in repressed promoter states.

By genome-wide biochemistry, this study reveals that a minimal set of information, comprising genomic DNA sequences, globular histones, and the molecular machinery of the remodeler, is sufficient to explain the placement and regulation of nucleosomes at their in vivo +1 positions for many promoters where appropriate DNA shape/mechanics signatures evolved. The identified mechanism of active information processing (Fig. 9b) provides allosteric control and versatile means for selective regulation, e.g., by epigenetic information such as histone modifications and variants as well as by the presence of sequence-specific factors such as transcription factors and pioneer factors. Signal integration of genome information from DNA shape/mechanics and sequence-specified GRF binding by the multi-subunit architecture of INO80 exemplifies such principles. In the accompanying study[39], we show how information from GRFs, DNA ends and positioned nucleosomes can be propagated into regular nucleosome arrays and how this process is regulated by remodeler rulers and nucleosome density. Collectively, this makes ATP-dependent remodelers the fundamental information processing hub for nucleosome positioning and thereby the primary architects of the first level of chromatin organization.

## Methods

**Embryonic *D. melanogaster* histone purification.** The preparation of embryonic *D. melanogaster* histones octamers was carried out as described before[30,68]. In brief, 50 g of 0–12 h old *D. melanogaster* embryos (strain OregonR) were dechorionated in 3% sodium hypochlorite, washed with dH₂0 and resuspended in 40 ml lysis buffer (15 mM K·HEPES pH 7.5, 10 mM KCl, 5 mM MgCl₂, 0.1 mM EDTA, 0.5 mM EGTA, 1 mM DTT, 0.2 mM PMSF, 10 % glycerol). Embryos were homogenized (Yamamoto homogenizer), filtered through cloth and centrifuged at 6500 × g for 15 min. Nuclei (brownish light pellet) were washed three times with 50 ml sucrose buffer (15 mM K·HEPES pH 7.5, 10 mM KCl, 5 mM MgCl₂, 0.05 mM EDTA, 0.25 mM EGTA, 1 mM DTT, 0.2 mM PMSF, 1.2% sucrose) and resuspended in 30 ml sucrose buffer containing 3 mM CaCl₂. To obtain mononucleosomes, nuclei were incubated for 10 min at 26 °C with 6250 units MNase (Sigma-Aldrich). Reaction was stopped with 10 mM EDTA, nuclei were pelleted and resuspended in 6 ml TE (10 mM Tris·HCl pH 7.6, 1 mM EDTA) containing 1 mM DTT and 0.2 mM PMSF followed by 30–45 min of rotation at 4 °C. Nuclei were centrifuged for 30 min at 15,300 × g at 4 °C. Solubilized mononucleosomes are found in the supernatant, which was applied to a pre-equilibrated hydroxyapatite column. After washing the hydroxyapatite column with 0.63 M KCl, histone octamers were eluted with 2 M KCl, concentrated and stored in 50% glycerol and 1x Complete (Roche) protease inhibitors without EDTA at −20 °C.

**Whole-genome plasmid library expansion.** The *S. cerevisiae* genomic plasmid library (pGP546) was originally described by Jones et al.[40] and purchased as a clonal glycerol stock collection from Open Biosystems. Library expansion was carried out via a Singer ROTOR plating machine (Singer Instruments) (8–12

rounds, three replicas). After 16 h, colonies were combined into $3 \times 2$ l of LB medium containing 50 µg/ml kanamycin and grown for 4 h. Cells were harvested and subjected to Plasmid Giga Preparation (PC 10,000 Kit, Macherey & Nagel).

**Salt gradient dialysis (SGD)**. For low, medium, and high assembly degrees, 10 µg of plasmid library DNA (*S. cerevisiae*) was mixed with ~2, 4, or 8 µg of *Drosophila* embryo histone octamers, respectively, in 100 µl assembly buffer (10 mM Tris·HCl, pH 7.6, 2 M NaCl, 1 mM EDTA, 0.05% IGEPAL CA630, 0.2 µg/µl BSA). For reconstitutions with precleaved DNA (Fig. 8), the plasmid library was digested with the respective RE and purified by phenol extraction/ethanol precipitation prior to SGD. Samples were transferred to Slide-A-lyzer mini dialysis devices, which were placed in a 3 l beaker containing 300 ml of high salt buffer (10 mM Tris·HCl pH 7.6, 2 M NaCl, 1 mM EDTA, 0.05% IGEPAL CA630, 14.3 mM β-mercaptoethanol), and dialyzed against a total of 3 l low salt buffer (10 mM Tris·HCl pH 7.6, 50 mM NaCl, 1 mM EDTA, 0.05% IGEPAL CA630, 1.4 mM β-mercaptoethanol) added continuously via a peristaltic pump over a time course of 16 h while stirring. β-mercaptoethanol was added freshly to all buffers. After complete transfer of low salt buffer, samples were dialyzed against 1 l low salt buffer for 1 h at room temperature. DNA concentration of the SGD chromatin preparations was estimated with a DS-11+ spektrophotometer (Denovix) and could be stored at 4 °C for several weeks. To estimate the extent of the assembly degree, an aliquot of the sample was subjected to MNase digestion (as described below) for MNase-ladder read out.

**Expression and purification of INO80 complex and respective mutants**. Coding sequences for *S. cerevisiae* Ino80 carrying a C-terminal 2xFlag-tag, Rvb1, Rvb2, Arp5-His, Ies6 (pFBDM_1) and Actin, Arp4, Arp8, Taf14, Ies2, Ies4, Ies1, Ies3, Ies5, and Nhp10 (pFBDM_2) were subcloned into pFBDM vectors[69] and sequence verified by Sanger sequencing. Bacmids of both vectors were generated using DH10 multibac cells[70]. Baculoviruses were generated in *Spodoptera frugiperda* (SF21) insect cells (IPLB-Sf21AE). *Trichoplusia ni* High Five (Hi5) insect cells (BTI-TN-5B1-4 Invitrogen) were co-infected with two baculoviruses 1/100 each. After 60 h cultivation at 27 °C, cells were harvested by centrifugation. For purification of the INO80 complex, cells were resuspended in lysis buffer (50 mM Tris·HCl pH 7.9, 500 mM NaCl, 10% glycerol, 1 mM DTT, SIGMAFAST™ protease inhibitor cocktail), sonified (Branson Sonifier, $3 \times 20$ s with 40% duty cycle and output control 3–4) and cleared by centrifugation (Sorvall Evolution RC, SS34 rotor, $15,000 \times g$). The supernatant was incubated for 1 h with 800 µl anti-Flag M2 Affinity Gel (product number A2220, Sigma-Aldrich) and centrifuged for 15 min at $1000 \times g$ and 4 °C. The anti-Flag resin was washed with buffer A (25 mM K·HEPES pH 8.0, 500 mM KCl, 10% glycerol, 0.025 mM IGEPAL CA630, 4 mM MgCl$_2$, 1 mM DTT) and buffer B (25 mM K·HEPES pH 8.0, 200 mM KCl, 10% glycerol, 0.02 mM IGEPAL CA630, 4 mM MgCl$_2$, 1 mM DTT). Recombinant INO80 complex was eluted with buffer B containing 0.22 mg/ml Flag Peptide (Sigma-Aldrich). Anion exchange chromatography (MonoQ 5/50 GL, GE Healthcare, Buffer: 25 mM K·HEPES pH 8.0, 4 mM MgCl$_2$, 1 mM DTT) using a linear KCl gradient 200–1000 mM) and, if required, size exclusion chromatography (Superose 6, 10/300 GL, 25 mM K·HEPES pH 8.0, 200 mM, 4 mM MgCl$_2$, 1 mM DTT) was used for further purification which resulted in a monodisperse INO80 complex (Fig. S1a, b, c). Using standard cloning techniques, three INO80 (2xFlag) HSA domain mutants[47] (HQ1, HQ2, HQ1/2; Figs. 2a and S1e), one N-terminal deletion mutant (Ino80$^{\Delta N}$, deletion of the first 461 amino acids of the N-terminus of Ino80) and two INO80 (2xFlag) Nhp10 module mutants ΔNhp10 (INO80 complex without Ies1, Ies3, Ies5, and Nhp10 but with Ino80 N-terminus) and HMGII (Figs. 2C and S1E) pFBDM vectors were generated and integrated into baculoviruses using MultiBac Technology as described above. Expression and purification of mutant INO80 complexes was essentially carried out as WT INO80 complex purification. The INO80 core complex from *Chaetomium thermophilum* (equivalent to the *S. cerevisiae* N-terminal deletion mutant) was essentially purified as described in[44].

**Genome-wide remodeling reaction**. All remodeling reactions were performed at 30 °C in 100 µl with final buffer conditions of 26.6 mM Na·HEPES pH 7.5, 1 mM Tris·HCl pH 7.6, 85.5 mM NaCl, 8 mM KCl, 10 mM ammonium sulfate, 10 mM creatine phosphate (Sigma-Aldrich), 3 mM MgCl$_2$, 2.5 mM ATP, 0.1 mM EDTA, 0.6 mM EGTA, 1 mM DTT, 14% glycerol, 20 ng/µl creatine kinase (Roche Applied Science). Remodeling reactions were started by adding 10 µl SGD chromatin corresponding to ~1 µg DNA assembled into nucleosomes and terminated by adding 0.8 units apyrase (NEB) followed by incubation at 30 °C for 30 min. Independent replicates of remodeling reactions refer to independent SGD chromatin preparations. The experimental conditions for each sample are detailed in Supplementary Data 1 and 2.

**MNase-seq**. After apyrase addition, remodeling reactions were supplemented with CaCl$_2$ to a final concentration of 1.5 mM and digested with 100 units MNase (Sigma) to generate mostly mononucleosomal DNA. In total, 10 mM EDTA and 0.5% SDS (final concentrations) were added to stop the MNase digest. After proteinase K treatment for 30 min at 37 °C, samples were ethanol precipitated and electrophoresed for 1.5–2 h at 100 V using a 1.5% agarose gel in 1x Tris-acetate-EDTA buffer. Mononucleosome bands were excised and purified with PureLink Quick Gel Extraction Kit (Thermo Fisher Scientific).

For library preparation, 10–50 ng of mononucleosomal DNA was incubated with 1.25 units Taq polymerase (NEB), 3.75 units T4 DNA polymerase (NEB), and 12.5 units T4-PNK (NEB) in 1x ligation buffer (B0202S, NEB) for 15 min at 12 °C, 15 min at 37 °C, and 20 min at 72 °C. To ligate NEBNext Multiplex Adaptors (0.75 µM final concentration, NEBNext Multiplex Oligos Kit) to the DNA, samples were incubated with T4 DNA ligase (NEB) at 25 °C for 15 min, followed by incubation with 2 units USER enzyme (NEB) for 10 min at 37 °C. Fragments were purified using 2 volumes AMPure XP beads (Beckman Coulter) and amplified for 8–10 cycles using NEBNext Multiplex Oligos, Phusion High-Fidelity DNA Polymerase (1 U, NEB), deoxynucleotide solution mix (dNTP, 2.5 mM, NEB), and Phusion HF Buffer (1x, NEB). The following protocol was applied for amplification: 98 °C for 30 s, 98 °C for 10 s, 65 °C for 30 s, 72 °C for 30 s with a final amplification step at 72 °C for 5 min. DNA content was assessed by using Qubit dsDNA HS Assay Kit (Invitrogen). PCR reactions were applied to an 1.5% agarose gel, needed fragment length (~270 bp) was excised and purified via PureLink Quick Gel Extraction Kit (Thermo Fisher Scientific). DNA was measured again with Qubit dsDNA HS Assay Kit and diluted to a final concentration of 10 nM (calculation based on the assumption that the DNA fragment length is 272 bp, i.e., 147 bp nucleosomal DNA and 122 bp sequencing adapter). Diluted samples were pooled according to sequencing reads (~6 Mio reads/sample). The final pool was quantified with BioAnalyzer (Agilent) and analyzed on an Illumina HiSeq 1500 in 50 bp single-end mode (Laboratory for Functional Genome Analysis, LAFUGA, LMU Munich).

**Expression and purification of human tailless histone octamers**. The genes for expression of tailless human histones H2A, H2B, and H4 were cloned in pET21b vectors (Merck, Darmstadt, Germany) by blunt-end ligation. The gene coding for human tailless H3 was cloned in a pETM-11 vector (kindly provided by EMBL, Heidelberg, Germany) carrying a N-terminal SUMO-tag by Gibson assembly[71]. The SUMO-tag was removed during octamer assembly. Constructs of tailless histones were designed according to globular domains identified by tryptic digest of full-length histone[72–74] and comprised the following amino acids: H2A: 13–118; H2B: 24–125; H3: 27–135; H4: 20–102. Histones were purified by a combination of inclusion body purification and ion-exchange chromatography, essentially as described previously[75,76]. In brief, histones were expressed in *E. coli* BL21 (DE3) cells (Merck, Darmstadt, Germany) for 2 h after induction with 1 mM IPTG at 37 °C and disrupted under non-denaturing conditions to separate inclusion bodies from lysate. Inclusion bodies were first washed with 1% Triton X-100. Subsequently, inclusion bodies were resuspended in 7 M guanidinium chloride and dialyzed against 8 M urea. Individual histones were purified by cation-exchange chromatography, refolded under low-salt conditions and polished by anion exchange chromatography. For long-time storage, histones were lyophilized overnight. For octamer reconstitution, histones were resuspended in 25 mM Tris, pH 7.5, 7 M guanidinium chloride, 0.25 mM DTT, mixed at 1.2-fold excess of H2A and H2B and dialyzed against 25 mM Tris·HCl pH 7.5, 2 M NaCl, 0.25 mM DTT overnight. In total, 1 mg/ml SENP2 protease was added after 3 h. The octamer of tailless histones was purified by size exclusion chromatography using a Superdex 200 16/60 column (GE Healthcare), which separated the octamer from aggregate, H2A/H2B dimers, the SENP2 protease, and the SUMO-tag. The purification was analyzed on a 18% polyacrylamide SDS gel stained with Coomassie. The octamer was concentrated to 3.0 mg/ml.

**Expression and purification of *S. cerevisiae* Reb1**. For genome-wide remodeling reaction *S. cerevisiae* Reb1 was purified exactly as described in[29]. For ATPase and mononucleosome sliding assays Reb1 was purified as follows: Reb1 was amplified from BY4741 genomic *S. cerevisiae* DNA by PCR (primers as in Supplementary Table 1) and cloned into pET21b (Novagen) via InFusion cloning (Clontech) with a Streptavidin tag at the C terminus. Correct sequences were verified via Sanger sequencing. Expression plasmids were transformed into BL21 (DE3) cd+ cells. Three liters of LB medium supplemented with 600 mg/l ampicillin were inoculated with 200 ml pre-culture. Cells were grown at 37 °C to an OD$_{600}$ of 0.6 (WPA CO8000 cell density meter). Induction was carried out by addition of IPTG to a final concentration of 1 mM. Cells were grown overnight at 18 °C, harvested by centrifugation (Sorvall Evolution RC SLC-6000 rotor, $2689 \times g$) and stored at −80 °C. Cells were resuspended in lysis buffer (50 mM Tris·HCl pH 7.9, 500 mM NaCl, 7% glycerol, 1 mM DTT, 7% sucrose and protease inhibitor 1:100), sonicated (Branson Sonifier 250, 5 min at 40–50% duty cycle and output control 4) and cleared by centrifugation (Sorvall Evolution RC, SS34 rotor, $15,000 \times g$). The supernatant was dialyzed over night against 2 l low salt buffer (25 mM K·HEPES pH 8.0, 50 mM KCl, 7% glycerol, 4 mM MgCl$_2$, 1 mM DTT). Heparin chromatography (5 ml column, elution buffer: 25 mM K·HEPES pH 8.0, 1 M KCl, 7% glycerol, 4 mM MgCl$_2$, 1 mM DTT) followed by size exclusion chromatography (Superdex 200 10/300, buffer: 25 mM K·HEPES pH 8.0, 200 mM KCl, 7% glycerol, 4 mM MgCl$_2$, 1 mM DTT) were used for purification. Peak fractions were analyzed by Coomassie SDS-PAGE. Fractions containing Reb1 were pooled, concentrated and stored at −80 °C.

**Preparation of mononucleosomes with recombinant human octamers**. Canonical human histones were provided by The Histone Source—Protein Expression and Purification Facility at Colorado State University. Lyophilized individual human histones were resuspended in 7 M guanidinium chloride, mixed at a 1.2-fold molar excess of H2A/H2B and dialyzed against 2 M NaCl for 16 h. Histone octamers were purified by size exclusion chromatography (HILoad 16/600 Superdex 200 column, GE Healthcare).

We used fluorescein-labeled Widom 601 DNA[56] with 80 bp extranucleosomal DNA (0N80 orientation) harboring an in vivo ChIP-Exo verified Reb1 binding site[77] of *S. cerevisiae* gene yGL167c (Reb1 binding motif: TTACCC) 64 or 84 bp distant to the 601 sequence. The DNA template (yGL267c_601) was amplified via PCR, purified by anion exchange chromatography (HiTrap DEAE FF, GE Healthcare) and vacuum concentrated. DNA and assembled histone octamer were mixed in 1.1-fold molar excess of DNA at 2 M NaCl. Over a time-period of 17 h at 4 °C the NaCl concentration was reduced to a final concentration of 50 mM NaCl. Again, anion exchange chromatography was used to purify reconstituted nucleosome core particle (NCP) which were then dialyzed to 50 mM NaCl. NCPs were concentrated to 1 mg/ml and stored at 4 °C.

**ATPase assay**. As described previously[47], we applied an NADH-based ATPase assay[78] to determine INO80's ATPase rate. 15 nM INO80 were incubated at 30 °C in a final volume of 50 μl assay buffer (25 mM K·HEPES pH 8.0, 50 mM KCl, 5 mM MgCl$_2$, 0.1 mg/ml BSA) with 0.5 mM phosphoenolpyruvate, 2 mM ATP, 0.2 mM NADH, and 25 units/ml lactate dehydrogenase/pyruvate kinase (Sigma-Aldrich) to monitor the NADH dependent fluorescence signal in non-binding, black, 384-well plates (Greiner) at an excitation wavelength of 340 nm and an emission wavelength of 460 nm over a 40-min period. We used the Tecan Infinite M1000 (Tecan) plate reader for read out. For all samples, ATPase activity was determined at maximum INO80 WT ATPase activity. ATPase activity was stimulated with 25 nM GL167c-0N80 mononucleosomes with or without equimolar ratios WT Reb1. Using maximal initial linear rates corrected for the buffer blank, we calculated final ATP turnover rates.

**Mononucleosome sliding assay**. Nucleosome sliding activity of INO80 wild type and mutant complexes were monitored on Reb1 site-0N80 mononucleosomes in absence and presence of 45 nM Reb1. INO80 at a concentration of 10 nM was incubated with 90 nM of Reb1 site-containing yGL167c-NCP601 and yGL167c-20-NCP601 mononucleosomes in sliding buffer at 26 °C (sliding buffer: 25 mM Na·HEPES pH 8.0, 60 mM KCl, 7% glycerol, 0.10 mg/ml BSA, 0.25 mM dithiothreitol and 2 mM MgCl$_2$). ATP and MgCl$_2$ at final concentrations of 1 and 2 mM, respectively, were added to start the sliding reaction. After 30, 60, 120, 300, 600, 1800, and 3600 s the reaction was stopped by adding lambda DNA (NEB) to a final concentration of 0.2 mg/ml. To separate distinct nucleosome species, we applied NativePAGE (NativePAGE Novex 4–16% Bis-Tris Protein Gels, Invitrogen). The fluorescein-labeled mononucleosomal DNA was visualized by an Typhoon™ FLA 9000 imager.

**Data processing**. Sequencing data were mapped to the *S. cerevisiae* SacCer3 (R64-1-1 build) genome using Bowtie[79]. Multiple matches were omitted. After mapping, data were imported into R Studio using GenomicAlignments[80]. Every read was shifted by 73 bp to cover the nucleosome dyad and extended to 50 bp. Genome coverage was calculated, and aligned to either in vivo +1 nucleosome positions[81], BamHI cut sites, Reb1 SLIM-ChIP hits[82], or Reb1 PWM hits[83]. Signal was normalized per gene in a 2001 bp window centered on the alignment point.

Heatmaps were sorted either by NDR length (distance between in vivo +1 and −1 nucleosome annotated by calling nucleosomes of in vivo MNase-seq data, see below) or by Reb1 binding score. For the latter, Reb1 SLIM-ChIP data (GSM2916407) were aligned to in vivo +1 nucleosome positions and sorted by signal strength in a 120 bp window 160 bp upstream of every +1 nucleosome.

For promotor grouping according to Reb1 site orientation, Reb1 SLIM-ChIP hits which contain a PWM site (±50 bp) and which are located within 400 bp upstream of in vivo +1 nucleosomes were used. Cluster 1 contains promotors where the Reb1 PWM motif is located on the sense strand and cluster 2, where the Reb1 PWM motif is located on the antisense strand. Cluster 3 contains Reb1 sites at bidirectional promotors.

**DNA shape and poly(dA)/(dT) analysis surrounding Reb1 binding sites**. The DNA sequence of the yeast genome (SacCer3) was downloaded from *Saccharomyces* Genome Database (SGD) and the DNA shape feature values (helix twist, propeller twist, minor groove width, and electrostatic potential) were calculated for the entire genome using the R package *DNAshapeR* (v1.10.0). Similar to[29], the resulting DNA shape vectors were smoothed with a 5-bp rollmean. For composite analysis, DNA shape feature specific values were extracted in a window of −2000 to 2000 bp around Reb1 binding sites, oriented with respect to Reb1 motif directionality, and averaged by base pair. Plotted distance around Reb1 features are indicated in respective figures.

For the poly(dA)/(dT) analysis, stretches of 6 nucleotide long polyA (5′-AAAAAA-3′) or polyT (5′-TTTTTT-3′) were identified in the yeast genome using R package *Biostrings* (v2.52.0). We assigned a coverage of one per nucleotide for

each hexa-homopolymeric stretch called. For composite analysis, poly(dA) or poly(dT) counts were extracted in a window of −2000 to 2000 bp around Reb1 binding sites, oriented with respect to Reb1 motif directionality, and averaged by base pair. Plotted distance around Reb1 features are indicated in respective figures.

**Annotation of +1 and -1 nucleosomes**. Nucleosome positions were called using the algorithm by Tirosh et al.[84]. Mononucleosomal fragments from log phase wt strain BY4741 in YPD medium (GEO deposition GSM4306342) were sequenced on an Illumina Genome analyzer, mapped to the SacCer3 genome with Bowtie[79] and read ends shifted by 73 bp downstream to derive dyad positions. Dyad-density counts were smoothed with sliding Gaussian filter (width = 100, mean = 0, SD = 25) and sorted by decreasing values. Iteratively, the position with the highest value was added to the list of nucleosome dyad centers and all values for positions within ±120 bp of the position with the highest value were removed. The top 90% of nucleosome dyad centers, by value, constituted the final list of nucleosome positions. Plus 1 nucleosome dyad positions were defined as the dyad positions nearest to a TSS[81] within a window of 0 to +500 bp from the TSS. Minus 1 nucleosome dyad positions were defined as the first nucleosome upstream of the +1 nucleosome.

**Genome-wide principal component and DNA shape analysis of nucleosomes**. For PCA and DNA shape analysis, mononucleosomes were sequenced in 50 bp paired-end mode on an Illumina HiSeq 1500. If not stated otherwise, functions were called with default parameters. Read pairs were aligned using Bowtie2 (2.2.9)[79] with options "-X 250 --no-discordant --no-mixed --no-unal". Only unique matches were kept, and orphaned mates removed. Nucleosomes were called on each sample using Bioconductor/nucleR (2.16.0) on nucleosomal fragments defined by paired reads as follows: fragments were processed with trimming to 40 bp around the dyads and their coverage was calculated. Noise was removed using FFT filtering with parameter pcKeepComp = 0.02 and peak detection was carried out with threshold 99%.

For each sample in an analysis set, sample-specific dyad positions obtained by nucleosome calling were enlarged to 20 bp and all positions were merged across the samples. Overlapping regions were joined. We excluded regions locating closer than 250 bp to tile borders and those residing in a region with high artifactual signals (chr III, 91,000–93,000 bp).

On this joint set of nucleosome dyads, we counted the number of overlapping fragments (reduced to their center position) for each sample. With $x$ being the number of counts of sample-specific fragment centers overlapping one dyad region of the joint set and sum($x$) being the sum of all counts across all dyad regions in the sample the data was normalized using the formula: normalized occupancy (dyad region) = log2((($x$/sum($x$)) × 1000) + 0.001). The resulting matrix was subjected to PCA. K-means clustering was applied to the resulting principal components to group nucleosomes based on similar occupancy patterns across sample conditions.

DNA shape features in windows of 320 bp around dyad positions were calculated with Bioconductor/DNAshapeR (version 1.14.0). DNA rigidity scores at each nucleotide position in windows of 320 bp around dyad positions were calculated as the length of the longest consecutive $A_nT_m$ ($n ≥ 0$, $m ≥ 0$ and $n + m ≥ 2$) sequence element that contains this position[85].

**Reporting summary**. Further information on research design is available in the Nature Research Reporting Summary linked to this article.

## Data availability
The data that support this study are available from the corresponding author upon reasonable request. Data that were generated in the course of this study have been deposited in the NCBI Gene Expression Omnibus with the accession number GSE145093 and GSE140614. Publicly available Reb1 SLIM-ChIP data are available with the accession number GSM2916407 and *S. cerevisiae* transcriptome data used for annotation of TSS are available from ArrayExpress [http://www.ebi.ac.uk/arrayexpress] under accession number E-TABM-590. The DNA sequence of the *Saccharomyces cerevisiae* SacCer3 genome (R64-1-1) were retrieved from iGenomes [https://support.illumina.com/sequencing/sequencing_software/igenome.html]. Source data are provided with this paper.

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

## Acknowledgements

We thank Stefan Krebs and Helmut Blum at the Laboratory for Functional Genome Analysis (LAFUGA, Gene Center, LMU München) for high throughput sequencing, and Sigurd Braun for access to and help with the plating robot. We thank Marianne Schwarz and Jens Michaelis for helpful discussions and earlier experiments using recombinant histones and endogenously purified INO80 not reported in this paper. This study was funded by the German Research Foundation (SFB1064 to P.K. and K.P.H. and Gottfried Wilhelm Leibniz-Prize to K.P.H.), the European Research Council (ERC Advanced Grant INO3D to K.P.H.), and the NIH (grant R35GM130376 to R.R.) and HFSP (grant RGP0021/2018 to R.R.). N.K. is supported by HFSP grant LT000631/2017-L.

## Author contributions

Conceptualization: E.O., N.K., T.S., R.R., K.P.H., P.K., and S.E.; Data curation: E.O., N.K., and V.N.; Formal analysis: E.O., N.K., V.N., Y.W., and T.S.; Funding acquisition, project administration, supervision: K.P.H., P.K., and S.E.; Investigation: E.O., N.K., V.N., K.S., M.M., Y.W., T.S., and S.E.; Methodology: E.O., N.K., V.N., K.S., R.R., T.S., P.K., and S.E.; Validation: E.O., N.K., V.N., K.S., Y.W., R.R., T.S., P.K., and S.E.; Visualization: E.O., N.K., V.N., Y.W., T.S., P.K., and S.E.; Writing original draft: E.O., N.K., P.K., and S.E.; Writing—review and editing: E.O., N.K., V.N., T.S., R.R., K.P.H., P.K., and S.E.

## Funding

## Competing interests

The authors declare no competing interests.
