## [Peer Review File · Nature Communications]

REVIEWER COMMENTS

Reviewer #1 (Remarks to the Author):

Oberbeckmann E and Krietenstein N et al.. performed comprehensive genome analysis using their genome-wide chromatin reconstitution system, which enabled the analysis of nucleosome positioning without PTM of histone tails or histone variants. I would like to appreciate their efforts and energy to achieve this study. However, I have serious concerns about their MNaseSeq. First, unfortunately, despite the purpose of clarifying the relative position of each other, the analysis using only mononucleosome fraction was performed. At least di-nucleosome size or more should be analyzed by paired-end sequencing. Second, most data they deposited to GEO was sequenced by single reads 50bp so repetitive sequences should have been excluded from their analysis, which suggests most genomic regions were not analyzed. I am very sorry that I cannot support their conclusion in this manuscript

Reviewer #2 (Remarks to the Author):

Review of the manuscript submitted by Oberbeckmann and colleagues, with the title "Genome information processing by the INO80 chromatin remodeler positions nucleosomes"

The authors use genome wide salt gradient reconstitution of chromatin and recombinant INO80 complexes to study the INO80 dependent nucleosome positioning mechanism next to boundaries, either represented by a transcription factor (Reb1) or nucleosomes positioned by double strand breaks. The authors suggest that a DNA shape signature is read by the INO80 complex to position nucleosomes, which is different from the DNA encoded histone octamer positioning signal. Reb1 as a boundary factor reduces the ATPase activity of the remodeler and defines nucleosome phasing. The authors identify the responsible Ino80 modules and shed light on +1 nucleosome positioning.

Comments

This study is very elegant and the experiments of high quality. In my opinion it provides important molecular data on the mechanism of nucleosome positioning by remodeling enzymes and how the defined +1 positioned nucleosomes are established.

When characterizing their SGD system, the authors state that neither histone posttranslational modifications, nor histone variants would affect the position of the +1 nucleosomes. If think the authors should withdraw that comment, as histone variant (H2A.Z) contributes to less than 10% of the total H2A pool in fly histones and I doubt that a 10% occupancy and differential nucleosome positioning at the +1 site can be monitored. In order to draw such conclusion, the SGD assembly has to be performed with recombinant H2A.Z octamers. I have the same skepticism regarding the posttranslational modifications. I think, as long it is not cleanly tested, the authors can not make this statement. There is still the possibility that H2A.Z containing nucleosomes may behave differently.

The authors expressed mutant forms of the INO80 complexes that exhibited overall positioning of the +1 nucleosomes (except for the inactive mutant as expected), albeit precise positioning is altered for the HQ1 and HQ2 mutants. Mutants shift nucleosomes to 5 or 10 bp further downstream, suggesting a role for the Arp8 module in determining nucleosome positioning. Could it also be that the relative architecture of the complex is changed by the mutations and the relative arrangement of the non-mutated modules on the linker DNA is shifted? Are nucleosomes at the newly presented position bound with higher affinity and present a more thermodynamic stable remodeling intermediate. Is it possible to determine DNA/nucleosome binding affinities for the nucleosomes at the different positions with the mutants?

The DNA shape analysis is really striking and suggests a role in nucleosome positioning. The data is correlative from many nucleosomes analyzed on the yeast genome reconstituted in vitro. However, a direct proof would strongly support the authors finding. I suggest to design two or three DNA fragments with the predicted shape constraints (and deviating controls) and to directly test the nucleosome positioning on this mononucleosomal DNA. Having such a defined system one could test in detail the molecular mechanism of INO80 dependent nucleosome positioning. Asking whether the shape of DNA changes binding affinities of the complexes towards nucleosomal start- and end-positions. How do the authors imagine the positioning of the nucleosomes. ATP hydrolysis is still going on (even if it is reduced in the presence of Reb1), are the nucleosome always moving forth and back? Is the MNase-Seq peak reflecting the major residence time at this position? Is a broadening of the peak (as observed with the mutants) suggesting that the nucleosomes are wiggling between close positions and these now rather represent nucleosomes with more similar residence times. What are the arguments of the authors.

In Figure 3a the authors are presenting the SGD and Ino80 nucleosome shape data. In my opinion the authors should not only present the shape of all SGD nucleosomes, but compare the nucleosome positions only that have been moved by Ino80. What was the shape at the starting position of the nucleosome and how did it change after movement. And the nucleosomes should be sorted by movement direction.

Next the authors tested the effects of barriers on nucleosome positioning, revealing an integrative read-out of DNA sequence, barrier and enzyme. Reb1 phasing is symmetrical and strikingly reduces the overall ATPase activity of the complex. Is Ino80 in its N-terminal part stably interacting with Reb1, being as well targeted to this site? Which subunit does interact with Reb1 and reduces the overall ATPase activity? Do the authors have experiments to address these important questions? Using the mononucleosomal assay they should be able to come up with answers and testing binding affinities towards the substrates.

This is a series of complex and very clean experiments, addressing two quite distinct aspects. First the +1 positioning and second the Reb1 dependent phasing. The individual parts still require experiments to address the mechanism, but surely are interdependent for regulating nucleosome positioning at the +1 site.

Reviewer #3 (Remarks to the Author):

In the manuscript Oberbeckmann, Krietenstein et al, the investigators utilize genomic and biochemical approaches to systematically investigate the nucleosome spacing activities of the INO80 chromatin remodeler. INO80 is an abundant evolutionarily conserved remodeler that plays roles in many DNA-templated processes, such as replication, repair and transcription. In vivo, the INO80 complex regulates a large proportion, nearly 15% of the *S.cerevisiae* transcripts, which are involved in cell growth and metabolic optimization. Therefore, the mechanism of INO80 nucleosome positioning is of broad interest to chromatin biology research communities.

Generally speaking, the specific determinants of nucleosome positioning by chromatin remodelers in vivo has remained somewhat elusive. This study tackles this important problem by utilizing an expansive collection of *S.cerevisiae* genomic DNA as an in vitro template to interrogate the DNA sequence and GRF dependency of INO80 nucleosome positioning. Additionally, informed by structural studies, they incorporate additional INO80 mutants to delineate the modules and subunits of INO80 that are important for nucleosome spacing.

The authors find that GRFs, specifically Reb1, and DNA ends function as barrier elements to establish nucleosome positioning that is further dictated by DNA shape. Both the Arp8 and Nhp10 modules are needed for proper nucleosome spacing and reading of the DNA shape. Results are largely in agreement with INO80 structural studies and in vivo ChIP assays.

The study is systematic and unbiased, as is its companion study Oberbeckmann, Niebauer et al., which compliments the results of this study. It is also well written and is expected to be of broad interest to the chromatin remodeling community. The results are significant and will likely have a lasting impact in the field. I have just minor comments below.

- Can the authors comment on the nature of the plasmid DNA templates used in this study? Is it expected that supercoiled plasmids will impart physical restraints on nucleosome assembly? Could this present some sequence bias in the results?
- Can the authors test (or describe previous results) that demonstrate the Nhp10 HMG mutant has diminished DNA binding activities?

We are grateful to all three Reviewers for spending their valuable time on both manuscripts and for providing insightful, stimulating and constructive criticism. This prompted us to provide additional experiments, analyses and clarifications, which considerably improved the two manuscripts.

- We provide additional validation for the impact of single-end versus paired-end sequencing, the quality of our genome wide mapping as well as the Reb1 site annotation by methylation-based genome footprinting.
- We performed band shift experiments that address the effects of INO80 point mutations on nucleosome binding by the INO80 complex
- We extended our DNA shape analyses towards the related but distinct aspect of DNA mechanics by employing a novel DNA rigidity score developed in the Rohs group (new co-authors Wang Y and Rohs R). Our findings underscore the relevance of the identified nucleosome positioning mechanism, in particular as it enables a comparison to the study of Basu et al. (now accepted “in principle” at Nature) in which DNA mechanics were experimentally measured on a genomic scale using a library-based DNA circularization assay.
- We clarified the possible role of histone marks, the importance of point mutations versus subunit deletions, the contribution of remodelers to the thermodynamic nucleosome positioning landscape, the role of plasmid topology, MNase digestion bias, the regulatory role of the Ino80 N-terminus in INO80-regulation by Reb1, and nucleosome positioning as the result of stable dynamic equilibrium.
- We added illustrated models that emphasizes the relevance of our in vitro findings for the in vivo situation and summarize our findings regarding the mechanism of +1 nucleosome positioning by INO80.

These changes are marked in the manuscript and detailed in our point-by-point response below. In the following we use **as shorthand “GI manuscript”** for the manuscript “Genome information processing by the INO80 chromatin remodeler positions nucleosomes” and **“ruler manuscript”** for the manuscript “Ruler elements in chromatin remodelers set nucleosome array spacing and phasing”.

Reviewer #1 (Remarks to the Author):

Oberbeckmann E and Krietenstein N et al. performed comprehensive genome analysis using their genome-wide chromatin reconstitution system, which enabled the analysis of nucleosome positioning without PTM of histone tails or histone variants. I would like to appreciate their efforts and energy to achieve this study. However, I have serious concerns about their MNaseSeq. First, unfortunately, despite the purpose of clarifying the relative position of each other, the analysis using only mononucleosome fraction was performed. At least di-nucleosome size or more should be analyzed by paired-end sequencing.

We appreciate the Reviewer's comment, but we respectfully disagree and like the Reviewer to consider the following: it is widely documented and **well-established practice to map nucleosome positions by sequencing mono-nucleosomal DNA fragments**, either in single- or in paired-end mode (e.g., (Albert et al. 2007; Jiang and Pugh 2009a; Quintales et al. 2015; Krietenstein et al. 2016; Ocampo et al. 2016) and see below). Mapping of dinucleosome positions is interesting in its own right, but neither necessary nor advantageous for mapping nucleosome positioning patterns from population averages of nucleosome positions. It was explicitly shown that **mapping dinucleosome MNase-fragments** by paired-end sequencing yields about the same patterns as for mono-nucleosome fragments (Ocampo et al. 2019). However, as the **+1 nucleosome position is largely underrepresented** in such datasets and as the +1 nucleosome position is especially relevant for our studies, **dinucleosome mapping is not suited** for our purposes. We quote here the respective sentence from Ocampo et al., 2019: **“Wild-type dinucleosomes show a fairly even occupancy similar to that of mononucleosomes, except for the +1 nucleosome, which is reduced, probably because the probability of MNase cutting upstream in the NDR is always high, increasing the probability of releasing the +1 nucleosome as a mononucleosome (Fig. 5A).”**, and show their respective Figure 5A below. Based on these reasons and experimental validation, we underscore that our well-established MNase-seq approach is valid to draw the conclusions presented in our study.

Second, most data they deposited to GEO was sequenced by single reads 50bp so not repetitive sequence should have been excluded from their analysis, which suggests most genomic regions were not analyzed. I am very sorry that I cannot support their conclusion in this manuscript

In our study, **we work with genomic sequences of *S. cerevisiae*, *S. pombe* and *E. coli***. The percentage of **non-unique repetitive sequences** in all these three genomes is **very minor** (7.5 %, 4.8% and 8%, respectively). We do exclude such non-unique sequences from our analyses as stated in the Methods section (p. 38 GI manuscript and p. 35 ruler manuscript: “Multiple matches were omitted.”), but **do retain the vast majority of genomic regions**. We wonder if the Reviewer's comment was prompted as he/she actually looked into our primary data and found that the

percentage of mappable reads for our in vitro reconstitution samples is around 50-60%. However, this does not reflect the percentage of genomic regions covered, but the percentage of sequencing reads mapped to the genomes. Our sequencing reads encompass not only regions from the three genomes, but also from the plasmid backbones as we use genomic plasmid libraries for in vitro reconstitution. Given the relative sizes of plasmid backbone versus average genomic insert, it is expected that 40-50% of the reads stem from the plasmid backbone and are not mapped to the respective genomes.

Finally, the **population average nucleosome mapping** as mostly used in our study **does not significantly benefit from paired-end versus single-end sequencing** of mono-nucleosomal DNA. We show below for the Reviewer's perusal the +1 nucleosome-aligned nucleosome patterns of three samples, for which we did use paired-end sequencing. The **average patterns** generated by either using both ends or only one end of the sequencing reads **are superimposable**. We used paired-end sequencing only in the context of our DNA shape analysis (Fig. 3-5, GI manuscript) for calling individual nucleosome dyads in contrast to population averages.

Reviewer #2 (Remarks to the Author):

Review of the manuscript submitted by Oberbeckmann and colleagues, with the title “Genome information processing by the INO80 chromatin remodeler positions nucleosomes”

The authors use genome wide salt gradient reconstitution of chromatin and recombinant INO80 complexes to study the INO80 dependent nucleosome positioning mechanism next to boundaries, either represented by a transcription factor (Reb1) or nucleosomes positioned by double strand breaks.

The authors suggest that a DNA shape signature is read by the INO80 complex to position nucleosomes, which is different from the DNA encoded histone octamer positioning signal. Reb1 as a boundary factor reduces the ATPase activity of the remodeler and defines nucleosome phasing. The authors identify the responsible Ino80 modules and shed light on +1 nucleosome positioning.

Comments

This study is very elegant and the experiments of high quality. In my opinion it provides important molecular data on the mechanism of nucleosome positioning by remodeling enzymes and how the defined +1 positioned nucleosomes are established.

We thank the Reviewer for the positive assessment of our study.

When characterizing their SGD system, the authors state that neither histone posttranslational modifications, nor histone variants would affect the position of the +1 nucleosomes. If think the authors should withdraw that comment, as histone variant (H2A.Z) contributes to less than 10% of the total H2A pool in fly histones and I doubt that a 10% occupancy and differential nucleosome positioning at the +1 site can be monitored. In order to draw such conclusion, the SGD assembly has to be performed with recombinant H2A.Z octamers. I have the same skepticism regarding the posttranslational modifications. I think, as long it is not cleanly tested, the authors can not make this statement. There is still the possibility that H2A.Z containing nucleosomes may behave differently.

We apologize for not clearly stating what we mean. We did not mean to say that histone variants and PTMs do not *affect* nucleosome positioning, but rather that they are not absolutely *required* for the principal activity of INO80 to generate the in vivo-like +1 nucleosome positions. We fully agree with the reviewer: histone variants and PTMs are expected to play a pivotal role in regulation of nucleosome positioning. Moreover, our whole-genome reconstitutions in conjunction with structural studies provide now an opportunity for future studies which will aim to dissect their precise impact and the underlying mechanism. **We now rephrased the respective section in the manuscript accordingly** (p. 8 GI manuscript: “Taken together, we concluded that neither histone modifications nor histone variants nor histone tails nor yeast-specific modifications are absolutely required for INO80 principal activity to position in vivo-like +1 nucleosome. Consequently, INO80 can generate such positioning solely by processing information from genomic DNA sequences and the globular histone octamer. Nonetheless, a readout of epigenetic information by remodelers is expected to play a pivotal role in the regulation of nucleosome positioning, e.g., in response to changes in the cellular environment, as discussed further below.”).

The authors expressed mutant forms of the INO80 complexes that exhibited overall positioning of the +1 nucleosomes (except for the inactive mutant as expected), albeit precise positioning is altered for the HQ1 and HQ2 mutants. Mutants shift nucleosomes to 5 or 10 bp further downstream, suggesting a role for the Arp8 module in determining nucleosome positioning. Could it also be that the relative architecture of the complex is changed by the mutations and the relative arrangement of the non-mutated modules on the linker DNA is shifted?

We thank the reviewer for this insightful comment. Many previous studies utilized endogenously purified deletion mutants of remodeler complexes that lacked subunits or entire modules and such mutations are indeed likely to alter the relative architecture of the complex. This motivated us in the first place to design **structure-based point mutations that do not interfere**, at least to our current knowledge of the INO80 structure, **with subunit-subunit interactions**. In addition, our purifications showed that wild type versus mutant INO80 apo complexes displayed a similar overall stability. To verify whether the structure-guided point mutations directly impact DNA binding of the INO80 complex, we performed a careful biochemical analysis. HQ1 and HQ2 mutants were biochemically assessed by using EMSAs as described in our recent publication (Knoll et al. 2018). In addition, we performed now similar experiments with INO80 complexes harboring point mutations of Nhp10. **These EMSAs show that Nhp10 HMG point mutations directly affect binding of nucleosomes that carry 80 bp linker DNA and we include these data now in Supplementary Fig. 6 of the ruler manuscript (see also answer to reviewer #3)**. Nevertheless, there remains the possibility that some of the deletion mutants alter the conformational flexibility of other parts of INO80. In particular, this might be the case for the N-terminal deletion mutant of Ino80. We speculate that the N-terminal region of Ino80 may fold back onto the Arp8-module, which may in turn affect the dynamics of the Arp8-module. **We include this interpretation now in our manuscript (p. 15 GI manuscript):** “The detailed mechanism of this intriguing allosteric communication across a distance of more than 70 bp

linker DNA awaits further structural studies. However, based on the regulatory role of the N-terminal region of Ino80, even in the absence of the Nhp10 module, we cautiously speculate that it might serve not only as a binding platform for Nhp10, but that it stimulates the activity of INO80 in absence of Reb1 possibly via restricting the dynamics of the Arp8 module.”

Ultimately, cryoEM studies are required to determine the entire conformational space of INO80 and to dissect the allosteric impact of the identified mutations at a detailed structural level. However, such studies are well beyond the scope of the current study.

Are nucleosomes at the newly presented position bound with higher affinity and present a more thermodynamic stable remodeling intermediate. Is it possible to determine DNA/nucleosome binding affinities for the nucleosomes at the different positions with the mutants?

This is an interesting question. However, remodeler-generated nucleosome positions do not mainly reflect binding affinities of the histone octamer to the DNA, but rather a threefold combination of the interactions between DNA, octamer and remodeler during the ATP dependent remodeler mechanism. The mere fact that different remodelers or remodeler mutants, despite working on the same octamers and DNA sequences, generate different nucleosome positions argues that **remodelers do not just kinetically “lubricate” the octamers to find their thermodynamically preferred positions** (otherwise different remodelers should generate the same positions), **but that the remodelers override intrinsic octamer preferences** (intrinsic thermodynamic binding affinities), i.e, remodelers themselves tweak the thermodynamic landscape and thereby co-determine the resulting positions. Our studies are based on this central argument, which we published earlier (Zhang et al. 2011; Korber 2012; Krietenstein et al. 2016), i.e., we wish to understand in which way remodelers contribute positioning information on top of the information inherent to the interactions between the histone octamer and the DNA sequence. We explicitly include this argument now (p. 19 GI manuscript. “The mere observation that INO80 and RSC remodelers generate different nucleosome positions, despite working on the same histone octamers and DNA sequences, suggested previously (Krietenstein et al. 2016), (Rippe et al. 2007) that remodelers do not just allow histone octamers to occupy their thermodynamically preferred positions (otherwise different remodelers would generate the same positions), but that remodelers, as revealed in this study for INO80, read genomic information, actively override octamer preferences and shape the positioning landscape in a remodeler-specific way.”). Therefore, and as direct answer to the Reviewer’s comment: As the ATP dependent remodeling reaction and remodeler binding itself contribute to resulting nucleosome stability, **there is currently no suitable method available that could compare relative thermodynamic stabilities of remodeler-generated nucleosome positions in the presence of the remodeler and its remodeling activity.** Note that nucleosomes are kinetically frozen under our physiological conditions in the absence of the remodeler. This means that the nucleosome will stay where the remodeler positioned it even in the absence of remodeling (see also below). However, the stability of the nucleosome at this position in the absence of the remodeler is not informative for the question how the nucleosome got there as the stability may be wholly different in the presence of the remodeler.

The DNA shape analysis is really striking and suggests a role in nucleosome positioning. The data is correlative from many nucleosomes analyzed on the yeast genome reconstituted in vitro. However, a direct proof would strongly support the authors finding. I suggest to design two or three DNA fragments with the predicted shape constraints (and deviating controls) and to directly test the nucleosome positioning on this mononucleosomal DNA. Having such a defined system one could test in detail the molecular mechanism of INO80 dependent nucleosome positioning. Asking whether the shape of DNA changes binding affinities of the complexes towards nucleosomal start- and end-positions.

We thank the Reviewer for this constructive suggestion. Such experiments were done in the context of the **study by Basu et al.**, which we co-submitted as relevant paper under consideration elsewhere. We refer here to their **Figure 2e,f** where **individual DNA sequences with favorable (less rigid linker) or unfavorable DNA (more rigid linker) mechanics features** were fused to 601 positioning sequences and then assessed for their INO80-induced sliding kinetics. Interestingly, Basu et al. performed their experiments also under saturating concentrations suggesting that the regulatory effect is not simply a difference in binding affinities. These results amount to the direct proof suggested by the Reviewer, i.e., that **DNA mechanics predictably affect INO80 nucleosome remodeling**.

Basu et al. determined experimentally DNA mechanics on a genome-wide scale via a library-based DNA circularization assay and their findings, obtained via this independent approach, closely matched our DNA shape analysis from whole genome reconstitutions. **This prompted us to now include a comparison between DNA shape and DNA mechanics by using a DNA rigidity score, which has been developed in the Remo Rohs laboratory (see new Fig. 4c and Supplementary Fig. 2a-c, GI manuscript), and to refer to the study by Basu et al (p. 12 GI manuscript: “This congruency immediately suggests a molecular mechanism by which an active readout not only through recognition of ground-state average DNA shape features, but also via ATP-hydrolysis driven perturbation of mechanical properties of DNA leads to the positioning of nucleosomes. The most immediate mechanical property of the double-helix is conformational flexibility. To assess this property on a genomic scale, we introduced a rigidity score that characterizes how rigid/flexible DNA is within a local region at bp resolution(Slattery et al. 2014). We considered A-tracts of consecutive ApA (TpT) or ApT bp steps as dominant factor in increasing rigidity due to strong stacking interactions combined with inter-bp hydrogen bonds in the major groove(Nelson et al. 1987; Rohs et al. 2009). The rigidity score accounts for the length of A-tracts as longer runs of ApA (TpT) and ApT steps without TpA steps or G/C bp increase rigidity of a DNA fragment. We observed that DNA rigidity is correlated with DNA shape features, and the correlation remains at a consistent level across INO80 positioned nucleosomes (Supplementary Fig. 2 a,b,c). This analysis reveals that +1 nucleosome positioning by INO80 involves placement of nucleosomes where DNA flexibility is increased at the -55 bp region between the ATPase motor and the Arp5 grip, while the promoter NDR region harbors a rigid DNA element where the Arp8-module is located (Fig. 4c). Intriguingly, a similarly rigid promoter DNA motif at the same distance in respect to the +1 nucleosome was also identified in a parallel study, where DNA mechanics were measured experimentally on a genomic scale via library-based DNA circularization assays (Basu et al. 2020).”**

The last sentence refers to Figure 3b of Basu et al. (see below), where they show a trough in intrinsic cyclizability that corresponds to a peak in DNA rigidity at the promoter NDR as shown in our Fig. 4c (GI manuscript).

How do the authors imagine the positioning of the nucleosomes. ATP hydrolysis is still going on (even if it is reduced in the presence of Reb1), are the nucleosome always moving forth and back?

The Reviewer's question aligns with our line of thinking. Our view is presented **in detail in the Discussion of the accompanying ruler manuscript**, there especially in **Fig. 7b-d**. We explicitly refer to this now in the Discussion (p. 21 GI manuscript: "Taken together, our findings lead to a model how regulation of nucleosome sliding direction bias upon interaction with a barrier can lead to stable nucleosome positioning and array formation (see Figure 7b-d (Oberbeckmann & Niebauer et al.))". Briefly, we propose that remodelers recognize some sort of alignment point (barrier protein, DNA end, neighboring nucleosome, DNA sequence element) such that sliding direction bias is regulated to generate a **stable dynamic equilibrium point**, which amounts to the resulting nucleosome position. Indeed, as wondered by the Reviewer, such positioning would not result in a static end product but in a dynamic equilibrium with remodeling going on. Nonetheless, and as noted above, nucleosomes are kinetically stuck without a remodeler under physiological conditions. Therefore, if a nucleosome is placed somewhere by a remodeler, it will stay there even after the remodeler is gone or inactive and as long as no other remodeler mobilizes it again. This is, also in response to the Reviewer's next question, the basis for all nucleosome mapping as this occurs in the absence of ATP and monitors a snap shot of nucleosome position distributions after nucleosome dynamics were stopped. An MNase-seq peak therefore reflects where most nucleosomes out of a nucleosome population are observed at a particular time point (population average). This is conceptually the same, but not experimentally obtained in MNase-seq, as the main probability where to observe a single nucleosome averaged over time (time average).

Is the MNase-Seq peak reflecting the major residence time at this position?

As explained above, the MNase-seq peak reflects the **population average at a given time point**.

Is a broadening of the peak (as observed with the mutants) suggesting that the nucleosomes are wiggling between close positions and these now rather represent nucleosomes with more similar residence times. What are the arguments of the authors.

The **broadening of the peak reflects a broadening of the distribution** of nucleosome positions relative to the alignment point in the population, i.e., fewer nucleosomes are positioned at or close to the alignment point and more nucleosomes are positioned further away. This concept is elaborated in a classical review on nucleosome mapping techniques by the Pugh group (Jiang and Pugh 2009b) and amounts to the colloquial difference between "well-positioned" versus "**fuzzy**" nucleosomes. In our study, this peak broadening shows that the DNA sequence underlying the in vivo +1 nucleosome positions has evolved such that it fits well the nucleosome positioning preferences of the WT INO80 complex. Conversely, this is why WT INO80 on its own positions nucleosomes at in vivo +1 positions (see also Fig. 2b, GI manuscript). In contrast, the mutant INO80 complexes have altered DNA sequence preferences (see also Fig. 5d GI manuscript) and position nucleosomes less precisely at in vivo +1 positions. We added now a sentence that explains better the significance of the peak broadening (p. 10 GI manuscript: "... ,which suggests that DNA sequences underlying in vivo +1 nucleosome positions correspond more to the DNA sequence preferences for nucleosome positioning of the WT versus the mutant INO80 complexes (see below).")

In Figure 3a the authors are presenting the SGD and Ino80 nucleosome shape data. In my opinion the authors should not only present the shape of all SGD nucleosomes, but compare the nucleosome positions only that have been moved by Ino80.

This relevant comment touches upon a discussion we previously had among ourselves, too. We decided to include nucleosomes that were positioned both by SGD and INO80 as a nucleosome that is not moved by INO80 still represents a nucleosome and therefore should still contain a sequence that reflects INO80's positioning preferences. Nonetheless, the point is somewhat moot as there were only 9 out of several thousand positions that were both in the SGD and the INO80 set.

Removing these 9 positions does not change the results as shown in the figure below.

What was the shape at the starting position of the nucleosome and how did it change after movement. And the nucleosomes should be sorted by movement direction.

This is an excellent suggestion. However, in our data **we cannot track the start and end positions for individual nucleosomes**. Further, as shown previously (e.g., (Zhang et al. 2009; Zhang et al. 2011)) SGD does not generate well-positioned nucleosomes, i.e., positions that would be occupied in the majority of molecules (sharp MNase-seq peak, see above) and could serve as "start positions". Nonetheless, our PCA/cluster analysis compares individual positions on individual molecules derived from paired-end sequencing and reveals common features, i.e. DNA shape features, among these positions that are different for SGD- versus INO80-nucleosomes. In this way one could say that **nucleosomes are moved from a DNA sequence with SGD- towards a DNA sequence with INO80-**

specific features. However, how far this movement was or in which direction along the genome, we cannot tell.

Next the authors tested the effects of barriers on nucleosome positioning, revealing an integrative read-out of DNA sequence, barrier and enzyme. Reb1 phasing is symmetrical and strikingly reduces the overall ATPase activity of the complex. Is Ino80 in its N-terminal part stably interacting with Reb1, being as well targeted to this site? Which subunit does interact with Reb1 and reduces the overall ATPase activity? Do the authors have experiments to address these important questions? Using the mono-nucleosomal assay they should be able to come up with answers and testing binding affinities towards the substrates.

As mentioned above, the Reviewer's constructive suggestions are spot-on with regard to our current work where we are addressing the mechanistic details of how INO80 senses and implements information input for nucleosome positioning. We do not have conclusive answers to these questions yet. Nonetheless, we presented already in the accompanying ruler manuscript ATPase assay data that address this point at least in part. These data (see figure below) are now moved from the accompanying ruler manuscript into the GI manuscript and presented in new Fig. 6e (GI manuscript). There we observed that **the regulating effect of Reb1 on the ATPase activity is gone if the N-terminus of Ino80 is either missing (Δ N-term mutant) or bound by a non-productive Nhp10 module (HMG point mutant).** We cautiously speculate that the interaction between Reb1 and INO80 works **via the Ino80 N-terminus** in a way that is regulated by Nhp10 module subunits. This may be linked to the discussion above about the interplay between the N-terminus of Ino80 and the Arp8 module.

This is a series of complex and very clean experiments, addressing two quite distinct aspects. First the +1 positioning and second the Reb1 dependent phasing. The individual parts still require experiments to address the mechanism, but surely are interdependent for regulating nucleosome positioning at the +1 site.

Reviewer #2 (Remarks to the Author):

Review of the manuscript by Oberbeckman and colleagues with the title “Ruler elements in chromatin remodelers set nucleosome spacing and phasing”

The authors do describe the mechanism of nucleosomes next next to a barrier and the subsequent spacing of neighboring nucleosomes. For this study they compared different remodeling enzymes and made use of an in vitro genome wide assay to analyze nucleosome positions. The authors located the so-called ruler element to the Nhp10 and Arp8 modules of the Ino80 complex.

The work is of high quality and addresses important questions in the field. The study is well designed and provides meaningful insights.

A few questions remain that should be addressed by the authors.

Reb1 binding is given in the figures as determined in vivo binding sites. Did the authors also determine binding of Reb1 to the reconstituted nucleosomal arrays? Does the in vivo binding reflect the binding in vitro? This would be important to know, as it sheds light on the importance of Reb1 mediated phasing.

The Reb1 binding can probably be seen by the sub-nucleosomal DNA being sequenced and analyzed, or by doing an in vitro CHIP.

We thank the Reviewer for this comment, which relates to an analysis that we did earlier but did not include in our manuscript. We address this now in new Supplementary Fig. 2a,b (ruler manuscript). Reb1 binding to free DNA was examined in the Pugh group by an in vitro version of their CHIP-exo approach called PB-exo (Rossi et al. 2018). **We compared in vitro Reb1 sites called via PB-exo with in vivo Reb1 sites called by SLIM-CHIP (Gutin et al. 2018)** and found a substantial overlap but also sites that were called only in vitro or only in vivo.

This raised the question that was also asked by the Reviewer, i.e., which Reb1 sites are bound in our in vitro chromatin reconstitution assays. Importantly, and as also implied in the Reviewer’s comment (“as it sheds light on the importance of Reb1 mediated phasing”), this question has two aspects. First, **which sites are bound by Reb1** in our assay, and, second, at which sites does bound Reb1 serve as a barrier for nucleosome positioning by a remodeler like INO80? We consider the second aspect

more important as Reb1 binding without effect on nucleosome positioning would not be relevant. Nonetheless, we addressed both aspects. First and in the context of a different study, we applied **DNA methylation footprinting** method (Krebs et al. 2017; Oberbeckmann et al. 2019) to reconstituted SGD chromatin + Reb1 and checked if we could detect Reb1 binding by protection from DNA methylation at Reb1 sites (plotted as occupancy = 1- methylation). We used low density (0.2) SGD chromatin to allow many potential Reb1 sites to be accessible. The potential Reb1 sites were differentiated according to the three groups of the above Venn diagram.

We observed clear Reb1 occupancy for Reb1 sites called by SLIM-ChIP in vivo but not for sites called only by PB-exo in vitro. This figure will be part of a different manuscript. Second, and this is now shown in the new Supplementary Figure 2b (ruler manuscript), **we compared the barrier effect of Reb1** for nucleosome positioning by INO80 at low, medium and high nucleosome density for the three Venn diagram groups of Reb1 sites.

The barrier effect of Reb1 was much more pronounced at sites called by SLIM-ChIP versus at sites called only by PB-exo. We conclude that **the in vivo called Reb1 sites are both bound and effective in terms of nucleosome barrier function in our reconstitution assays**, whereas sites called by PB-exo only probably reflect crosslinking of Reb1 to DNA that was detected by PB-exo but did not correspond to stable binding and was mostly not functionally relevant in our assay. This conclusion is now reflected in the new Supplementary Figure 2a,b (ruler manuscript) and in the main text (p. 7 ruler manuscript, “We chose Reb1 sites called in vivo by SLIM-ChIP (Gutin et al. 2018) as we saw strong effects upon alignment at these sites in contrast to sites called only for Reb1 binding to free DNA in vitro by PB-exo (Rossi et al. 2018) (Supplementary Fig. 2a,b)”.) and directly addresses the Reviewer’s question, i.e., in vivo Reb1 binding also reflects in vitro Reb1 binding in our assay, and not just binding, but also barrier functionality for nucleosome positioning.

How do the MNase digestion patterns of the low to high assembly states look alike? The

center of the peaks around the NFR are closer to the NFR in high MNase than in lower MNase conditions. If phasing would depend on Reb1 only the distances should be the same. Can it be that MNase digestions at low assembly ratios does generate a higher fraction of fragments smaller than 147bp? Agarose gel is given, but the size distribution after sequencing is not.

We wondered about this ourselves and found that this concern is not warranted. First, **if different digestion degrees at different assembly degrees significantly affected nucleosome positions, then we could not observe the striking consistency of nucleosome positions generated by Chd1** (Figs. 1d, 2c-e, ruler manuscript) across the whole range of assembly degrees. This argument is now incorporated in the manuscript (p. 7 ruler manuscript, „The observation that nucleosome phasing and spacing was largely invariant for Chd1 samples at all densities excluded that respective differences for the other remodelers were due to MNase digestion degrees varying with nucleosome densities.“). Second, and as requested by the Reviewer, plotting the **size distributions of sequenced DNA fragments after paired-end sequencing shows only very minor differences for different assembly degrees** (see new plots below that show two examples of fragment size distributions at the three different assembly degrees (low, medium, high)).

It would be good to see the statistical variation between the individual repeats in the plot. Variation should be plotted in the Figures to assess robustness of the results.

We appreciate the Reviewer's comment and **did take great care to show variation** between individual repeats already in our first paper version. First, we plotted data points for all individual replicates in Figs. 2c-e, 3b,c, 5b-e and 6b-d (ruler manuscript). Sometimes in Figs. 2d, 3c, 5c and 6c all data points fall within the dimension of the box stating the average value. Second, we plotted the nucleosome profiles of all individual replicates in the Supplementary Figures 1b,c, 2c,d, 3d, 4a and 5a (ruler manuscript). We tried combining all replicates into an average plot with an error bar ribbon (see below), but consider showing the individual replicates as more transparent and telling and also more in agreement with Nature Journal editorial policies (see nr-editorial-policy checklist) with regard to showing individual data points for replicates.

data from Supplementary Fig. 1b (ruler manuscript)
 individual replicates plotted plotted with error bar ribbons

Minor points

Page 9 first paragraph. It must read “For INO80, linker lengths steadily decreased ...

No, the original wording is correct. Linkers do become longer with decreasing nucleosome density.

Reviewer #3 (Remarks to the Author):

In the study Oberbeckmann, Niebauer et al., that authors utilize genomic and biochemical approaches to systematically investigate the nucleosome spacing activities of the several chromatin remodelers. Chromatin remodelers play essential roles in DNA-templated processes, including transcription, replication and repair. In vivo investigations of chromatin remodelers demonstrate that they are essential for nucleosome positioning and phasing. However, how chromatin remodelers facilitate their distinct activities in vivo remains largely unknown.

This study tackles this important problem by utilizing an expansive collection of *S.cerevisiae* genomic DNA as an in vitro template to interrogate the determinants of all chromatin remodelers with nucleosome spacing activity. Additionally, informed by structural studies, they incorporate additional INO80 mutants to delineate the modules and subunits of INO80 that are important for nucleosome spacing.

The authors find that INO80, ISW2, ISW1a and Chd1, but not Fun30, align nucleosome arrays symmetrically around GRF barrier, specifically Reb1, and also DNA break sites. The distance to the barrier is specific to each remodeler as well as nucleosome density. These remodeler-specific nucleosome rulers establish spacing that are either independent or depending on nucleosome density. The authors make note of the companion paper Oberbeckmann, Krietenstein et al., that identifies DNA sequence specificity for chromatin remodelers, and also finds that eukaryotic sequences adapt arrays with better efficiency than prokaryotic DNA.

The authors then interrogate the INO80 complex for modules and subunits that dictate nucleosome spacing. Using mutant complexes they identified different ruler mechanisms attributed to the Arp8 and Nhp10 modules, which are not entirely coupled to ATPase activity. Lastly, because of the non-conserved Nhp10 module, they postulate both species-specific and conserved ruler mechanisms.

The study is systematic and unbiased, as is its companion study Oberbeckmann, Krietenstein et al., which compliments the results of this study. In general, the manuscript is extremely well written, although at times lengthy (for example, as is the Discussion). It is also expansive in the use of important variables and multiple observations, such as nucleosome density, linker length, nucleosome positioning, and anchor mechanisms. In this regard, the study is extremely thorough, yet also very dense.

We thank the Reviewer for the appreciation of our work.

The multitude of detailed in vitro results is a bit difficult to assimilate in the context of in vivo observations. Perhaps the authors could present an illustrated model or table that incorporates the main findings of each remodeler with an emphasis on results that correlate with in vivo observations?

We thank the Reviewer for this comment and provide now **both an illustrated model as well as a table with a summary** of main in vitro results that correlate with in vivo observations in the new Fig. 7a (ruler manuscript).

Minor points:

- Can the authors comment on the nature of the plasmid DNA templates used in this study? Is it expected that supercoiled plasmids will impart physical restraints on nucleosome assembly? Could this present some sequence bias in the results?

We thank the Reviewer for this knowledgeable comment. The **role of plasmid topology was indeed investigated by us while establishing our *in vitro* reconstitution approach**. The plasmids during salt gradient dialysis (SGD) reconstitution are circular and mostly supercoiled. However, once the nucleosomes are reconstituted during SGD and this chromatin is pipetted out of the dialysis tube, the plasmids are mostly nicked after SGD, probably due to mechanical shearing as published by us (Krietenstein et al. 2012). This means that nucleosome remodeling by the remodelers, which always takes place after SGD, is not constrained by topological strains as wondered by the reviewer. As direct demonstration, we compare in the plot below the INO80-generated nucleosome positioning in promoter regions in samples with and without included BamHI restriction enzyme. BamHI was included to show the INO80-generated arrays at dsDNA breaks (e.g., Fig. 2a, ruler manuscript), but here we use the fact that BamHI also linearized the plasmids to demonstrate for the Reviewer that the **patterns are highly similar for linear versus circular plasmids**.

- Can the authors test (or describe previous results) that demonstrate the Nhp10 HMG mutant has diminished DNA binding activities?

This is a very valid point. Previously (Tosi et al. 2013), we showed that an INO80 complex lacking the Nhp10 module had reduced binding affinity for mononucleosomes. Now, in response to the Reviewer's question, **we did band shift assays also for the HMGII point mutant and found that also this mutant showed reduced binding affinity for mononucleosomes.** We present these results now on p. 13/24 (ruler manuscript, "Third, the Nhp10 module subunits contributed to the ruler mainly through the HMG box of Nhp10 as the respective point mutations (HMGII mutant) mimicked the effects upon lack of all Nhp10 module subunits (Δ Nhp10 mutant) (Fig. 6c,d). This was also true with regard to reduced nucleosome binding affinity, which was published for the Δ Nhp10 mutant (Tosi et al. 2013) and also found here for the HMGII mutant (Supplementary Fig. 6).") and in the new Supplementary Fig. 6 (ruler manuscript).

Supplementary Figure 6. **Reduced affinity for mononucleosome binding by the HMGII versus the WT INO80 complex.** Two replicates of gel retardation assays with the indicated concentrations of recombinant WT or HMGII mutant INO80 complexes and 20 nM mononucleosomes where a nucleosome reconstituted on the Widom 601 positioning sequence was flanked by no linker DNA on one side and 80 bp linker DNA on the other side (INO80 mononucleosome).

References:

- Albert I, Mavrich TN, Tomsho LP, Qi J, Zanton SJ, Schuster SC, Pugh BF. 2007. Translational and rotational settings of H2A.Z nucleosomes across the *Saccharomyces cerevisiae* genome. *Nature* **446**: 572-576.
- Basu A, Bobrovnikov DG, Qureshi Z, Kayikcioglu T, Ngo TTM, Ranjan A, Eustermann S, Cieza B, Morgan MT, Hejna M et al. 2020. Measuring DNA mechanics on the genome scale. doi:10.1101/2020.08.17.255042 %J bioRxiv: 2020.2008.2017.255042.
- Gutin J, Sadeh R, Bodenheimer N, Joseph-Strauss D, Klein-Brill A, Alajem A, Ram O, Friedman N. 2018. Fine-Resolution Mapping of TF Binding and Chromatin Interactions. *Cell reports* **22**: 2797-2807.
- Jiang C, Pugh BF. 2009a. A compiled and systematic reference map of nucleosome positions across the *Saccharomyces cerevisiae* genome. *Genome biology* **10**: R109.
- Jiang C, Pugh BF. 2009b. Nucleosome positioning and gene regulation: advances through genomics. *Nature reviews Genetics* **10**: 161-172.
- Knoll KR, Eustermann S, Niebauer V, Oberbeckmann E, Stoehr G, Schall K, Tosi A, Schwarz M, Buchfellner A, Korber P et al. 2018. The nuclear actin-containing Arp8 module is a linker DNA sensor driving INO80 chromatin remodeling. *Nature structural & molecular biology* **25**: 823-832.
- Korber P. 2012. Active nucleosome positioning beyond intrinsic biophysics is revealed by in vitro reconstitution. *Biochemical Society transactions* **40**: 377-382.
- Krebs AR, Imanci D, Hoerner L, Gaidatzis D, Burger L, Schubeler D. 2017. Genome-wide Single-Molecule Footprinting Reveals High RNA Polymerase II Turnover at Paused Promoters. *Molecular cell* **67**: 411-422.e414.
- Krietenstein N, Wal M, Watanabe S, Park B, Peterson CL, Pugh BF, Korber P. 2016. Genomic Nucleosome Organization Reconstituted with Pure Proteins. *Cell* **167**: 709-721.e712.
- Krietenstein N, Wippo CJ, Lieleg C, Korber P. 2012. Genome-wide in vitro reconstitution of yeast chromatin with in vivo-like nucleosome positioning. *Methods in enzymology* **513**: 205-232.
- Nelson HCM, Finch JT, Luisi BF, Klug A. 1987. The Structure of an Oligo(Da).Oligo(Dt) Tract and Its Biological Implications. *Nature* **330**: 221-226.
- Oberbeckmann E, Wolff M, Krietenstein N, Heron M, Ellins JL, Schmid A, Krebs S, Blum H, Gerland U, Korber P. 2019. Absolute nucleosome occupancy map for the *Saccharomyces cerevisiae* genome. *Genome research* **29**: 1996-2009.
- Ocampo J, Chereji RV, Eriksson PR, Clark DJ. 2016. The ISW1 and CHD1 ATP-dependent chromatin remodelers compete to set nucleosome spacing in vivo. *Nucleic acids research* **44**: 4625-4635.
- Ocampo J, Chereji RV, Eriksson PR, Clark DJ. 2019. Contrasting roles of the RSC and ISW1/CHD1 chromatin remodelers in RNA polymerase II elongation and termination. *Genome research* **29**: 407-417.
- Quintales L, Vazquez E, Antequera F. 2015. Comparative analysis of methods for genome-wide nucleosome cartography. *Briefings in bioinformatics* **16**: 576-587.
- Rippe K, Schrader A, Riede P, Strohn R, Lehmann E, Langst G. 2007. DNA sequence- and conformation-directed positioning of nucleosomes by chromatin-remodeling complexes. *Proceedings of the National Academy of Sciences of the United States of America* **104**: 15635-15640.
- Rohs R, West SM, Liu P, Honig B. 2009. Nuance in the double-helix and its role in protein-DNA recognition. *Curr Opin Struc Biol* **19**: 171-177.
- Rossi MJ, Lai WKM, Pugh BF. 2018. Genome-wide determinants of sequence-specific DNA binding of general regulatory factors. *Genome research* **28**: 497-508.

- Slattery M, Zhou TY, Yang L, Machado ACD, Gordan R, Rohs R. 2014. Absence of a simple code: how transcription factors read the genome. *Trends Biochem Sci* **39**: 381-399.
- Tosi A, Haas C, Herzog F, Gilmozzi A, Berninghausen O, Ungewickell C, Gerhold CB, Lakomek K, Aebersold R, Beckmann R et al. 2013. Structure and subunit topology of the INO80 chromatin remodeler and its nucleosome complex. *Cell* **154**: 1207-1219.
- Zhang Y, Moqtaderi Z, Rattner BP, Euskirchen G, Snyder M, Kadonaga JT, Liu XS, Struhl K. 2009. Intrinsic histone-DNA interactions are not the major determinant of nucleosome positions in vivo. *Nature structural & molecular biology* **16**: 847-852.
- Zhang Z, Wippo CJ, Wal M, Ward E, Korber P, Pugh BF. 2011. A packing mechanism for nucleosome organization reconstituted across a eukaryotic genome. *Science (New York, NY)* **332**: 977-980.

REVIEWERS' COMMENTS

Reviewer #1 (Remarks to the Author):

First of all, I would like to thank the authors for their careful efforts to explain their work. In particular, I was satisfied with the explanation of the genomic repeatable regions.

On the other hand, they do not seem to understand my intention that they should analyze with paired-end data; the reason for analyzing with PE does not imply an increase in mapping rates or the need to sequence only di-nucleosomes. In this study, which is concerned with the relative position of nucleosomes, it is important to analyze the various fragments that are assumed to exist at both ends of the nucleosome to determine the exact fragmentation size in order to evaluate their "co-occurrence". I am also disappointed that they tried to clear this point without additional experiments. I think that a careful design of the experiment is desirable if authors want to make a major conceptual proposal, even if authors have to divide it into two papers.

Reviewer #2 (Remarks to the Author):

The authors did sufficiently address the concern of the reviewer with explanations and additional data analysis. In my opinion the manuscripts are suitable for publication in Nature Communications.

Reviewer #4 (Remarks to the Author):

In the two accompanying manuscripts, Oberbeckmann et al. performed a systematic and comparative analysis of nucleosome positioning near gene promoters by ATP-dependent chromatin remodelers and the effects of genomic sequences and DNA shape, TF-barriers, DNA-end barriers, as well as nucleosome densities. The authors used chromatin reconstituted on whole-genome-derived DNA templates with native and recombinant histones, purified/reconstituted yeast chromatin remodeling complexes, and the TF Reb1 - allowing the determination of individual/combined effects of these factors. Analysis of the resulting nucleosome positions by MNase-seq showed remodeler-specific nucleosome positioning patterns. The authors further focused on the INO80 chromatin remodeler and showed how its Nhp10 and Arp8 modules act as molecular rulers to position the promoter-proximal +1 nucleosomes based on distance from a TF-barrier or DNA shape features.

Taken together, the two manuscripts provide novel and important conceptual advancement towards some central questions in chromatin organization and transcription regulation – how are well-organized arrays of nucleosomes flanking active-gene promoters are generated, and what are the determinants by which ATP-dependent remodelers position nucleosomes at these genomic regions. I enthusiastically support the publication of the manuscripts without additional experiments/analyses.